# LOGICAL ACTIVATION FUNCTIONS: LOGIT-SPACE EQUIVALENTS OF BOOLEAN OPERATORS

## ABSTRACT

The choice of activation functions and their motivation is a long-standing issue within the neural network community. Neuronal representations within artificial neural networks are commonly understood as logits, representing the log-odds score of presence (versus absence) of features within the stimulus. In this work, we investigate the implications on activation functions of considering features to be logits. We derive operations equivalent to AND, OR, and XNOR for log-odds ratio representations of independent probabilities. Since these functions involve taking multiple exponents and logarithms, they are not well suited to be directly used within neural networks. Consequently, we construct efficient approximations named $\text{AND}_{\text{AIL}}$ (the AND operator Approximate for Independent Logits), $\text{OR}_{\text{AIL}}$, and $\text{XNOR}_{\text{AIL}}$, which utilize only comparison and addition operations and can be deployed as activation functions in neural networks. Like MaxOut, $\text{AND}_{\text{AIL}}$ and $\text{OR}_{\text{AIL}}$ are generalizations of ReLU to two-dimensions. We deploy these new activation functions, both in isolation and in conjunction, and demonstrate their effectiveness on a variety of tasks including image classification, transfer learning, abstract reasoning, and compositional zero-shot learning.

## 1 INTRODUCTION

An activation function is a non-linearity which is interlaced between linear layers within an artificial neural network. The non-linearity is essential in order for higher-order representations to form, since otherwise the network would be degeneratively equivalent to a single linear layer.

Early artificial neural networks were inspired by biological neural networks, with the activation function analogous to a neuron's need to exceed a potentiation threshold in order to fire an action potential. Biological neurons are have long been known to be more complex than this simple abstraction, including features such as non-linearities in dendritic integration. Recent work has demonstrated that a single biological neuron can compute the XOR of its inputs (Gidon et al., 2020), a property long known to be lacking in artificial neurons (Minsky & Papert, 1969). This suggests that gains in artificial neurons can be made by using activation functions which operate on more than one input to the neuron at once.

The earliest artificial neural networks featured either logistic-sigmoid or tanh as their activation function. These activation functions were motivated by the idea that each layer of the network is building another layer of abstraction of the stimulus space from the last layer. Each neuron in a layer identifies whether certain properties or features are present within the stimulus, and the pre-activation (potentiation) value of the neuron indicates a score or logit for the presence of that feature. The sigmoid function, $\sigma(x) = 1/(1+e^{-x})$, was hence a natural choice of activation function, since as with logistic regression, this will convert the logits of features into probabilities.

There is some evidence that this interpretation has merit. Previous work has been done to identify which features neurons are tuned to. Examples include LSTM neurons tracking quotation marks, line length, and brackets (Karpathy et al., 2015); LSTM neurons tracking sentiment (Radford et al., 2017); methods for projecting features back to the input space to view them (Olah et al., 2017); and interpretable combinations of neural activities (Olah et al., 2020). Analogously, within biological neural networks, neurons are tuned to respond more strongly to certain stimuli, and less strongly to

others. At high representation levels, concept cells respond predominantly to certain concepts, such one's grandmother, or Jennifer Aniston (Gross, 2002; Quian Quiroga et al., 2005).

Sigmoidal activation functions are no longer commonly used within machine learning between layers of representations, though sigmoid is still widely used for gating operations which scale the magnitude of other features in an attention-like manner. The primary disadvantage of the sigmoid activation function is its vanishing gradient — as the potentiation rises, activity converges to a plateau, and hence the gradient goes to zero. This prevents feedback information propagating back through the network from the loss function to the early layers of the network, which consequently prevents it from learning to complete the task.

The Rectified Linear Unit activation function (Fukushima, 1980; Jarrett et al., 2009; Nair & Hinton, 2010), $\text{ReLU}(x) = \max(0, x)$, does not have this problem, since in its non-zero regime it has a gradient of 1. Another advantage of ReLU is it has very low computational demands. Since it is both effective and efficient, it has proven to be a highly choice of popular activation function. The chief drawback to ReLU is it has no sensitivity to changes across half of its input domain, and on average passes no gradient back to its inputs. This can lead to neurons dying[1] if their weights make them never reach their activation threshold. If sigmoid is equivalent to converting our feature logits into probabilities, then the ReLU activation function is equivalent to truncating our logits and denying any evidence for the absence of a feature. We hypothesise that omitting negative evidence in this way is undesirable and reduces the generalisation ability of ReLU-based networks.

Variants of ReLU have emerged, aiming to smooth out its transition between domains and provide a gradient in its inactive regime. These include ELU (Clevert et al., 2016), CELU (Barron, 2017), SELU (Klambauer et al., 2017), GELU (Hendrycks & Gimpel, 2020), SiLU (Elfwing et al., 2017; Ramachandran et al., 2017), and Mish (Misra, 2019). However, all these activation functions still bear the general shape of ReLU and truncate negative logits.

Fuzzy logic operators are generalizations of boolean logic operations to continuous variables, using rules similar to applying logical operators in probability space. Previous work has explored networks of fuzzy logic operations, including neural networks which use an activation functions that constitutes a learnable interpolation between fuzzy logic operators (Godfrey & Gashler, 2017). In this work, we introduce activation functions which are similar to fuzzy logic operators, but derived for working in logit space instead of in probability space.

In this work we set out to develop activation functions based on the principle that neurons encode logits — scores that represent the presence of features in the log-odds space. In Section 2 we derive and define these functions in detail for different logical operators, and then consider their performance on numerous task types including parity (Section 3.1), image classification (Sections 3.4 and 3.5), transfer learning (Section 3.6), abstract reasoning (Appendix A.14), soft-rule guided classification as exemplified by the Bach chorale dataset (Section 3.3), and compositional zero-shot learning (Appendix A.15). These tasks were selected to (1) survey the performance of the new activations on existing benchmark tasks, and (2) evaluate their performance on tasks which we suspect in particular may require logical reasoning and hence benefit from activation functions which apply these logical operations to logits.

## 2 DERIVATION

Manipulation of probabilities in logit-space is known to be more efficient for many calculations. For instance, the log-odds form of Bayes' Rule (Equation 9) states that the posterior logit equals the prior logit plus the log of the likelihood ratio for the new evidence (the log of the Bayes factor).Thus, working in logit-space allows us to perform Bayesian updates on many sources of evidence simultaneously, merely by summing together the log-likelihood ratios for the evidence. A weighted sum may be used if the amount of credence given to the sources differs — and this is precisely the operation performed by a linear layer in a neural network.

When considering sets of probabilities, a natural choice of operation to perform is measuring the joint probability of two events both occurring — the AND operation for probabilities. Suppose our input space is $x \in [0, 1]^2$, and the goal is to output $y > 0$ if $x_i = 1 \,\forall\, i$, and $y < 0$ otherwise, using model

---

[1]Though this problem is very rare when using BatchNorm to stabilise feature distributions.

with a weight vector $w$ and bias term $b$, such that $y = w^T x + b$. This can be trivially solved with the weight matrix $w = [1, 1]$ and bias term $b = -1.5$. However, since this is only a linear separator, the solution can not generalise to the case $y > 0$ iff $x_i > 0 \, \forall \, i$.

Similarly, let us consider how the OR function solved with a linear layer. Our goal is to output $y > 0$ if $\exists \, x_i = 1$, and $y < 0$ otherwise. The binary case can be trivially solved with the weight matrix $w = [1, 1]$ and bias term $b = -0.5$. The difference between this and the solution for AND is only an offset to our bias term. In each case, if the input space is expanded beyond binary to $\mathbb{R}^2$, the output can be violated by changing only one of the arguments.

## 2.1 AND

Suppose we are given $x$ and $y$ as the logits for the presence (vs absence) of two events, $X$ and $Y$. These logits have equivalent probability values, which can be obtained using the sigmoid function, $\sigma(u) = (1 + e^{-u})^{-1}$. Let us assume that the events $X$ and $Y$ are independent of each other. In this case, the probability of both events occurring (the joint probability) is $\mathrm{P}(X, Y) = \mathrm{P}(X \wedge Y) = \mathrm{P}(X) \, \mathrm{P}(Y) = \sigma(x) \, \sigma(y)$.

However, we wish to remain in logit-space, and must determine the logit of the joint probability, $\mathrm{logit}(\mathrm{P}(X, Y))$. This is given by

$$\mathrm{AND_{IL}} := \mathrm{logit}(\mathrm{P}(X \wedge Y)_{x \perp\!\!\!\perp y}) = \log\left(\frac{p}{1 - p}\right), \text{ where } p = \sigma(x) \, \sigma(y),$$

$$= \log\left(\frac{\sigma(x) \, \sigma(y)}{1 - \sigma(x) \, \sigma(y)}\right), \tag{1}$$

which we coin as $\mathrm{AND_{IL}}$, the AND operator for independent logits (IL). This 2d function is illustrated as a contour plot (Figure 1, left). Across the plane, the order of magnitude of the output is the same as at least one of the two inputs, scaling approximately linearly.

The approximately linear behaviour of the function is suitable for use as an activation function (no vanishing gradient), however taking exponents and logs scales poorly from a computational perspective. Hence, we developed a computationally efficient approximation as follows. Observe that we can loosely approximate $\mathrm{AND_{IL}}$ with the minimum function (Figure 1, right panel). This is equivalent to assuming the probability of **both** $X$ and $Y$ being true equals the probability of the **least likely** of $X$ and $Y$ being true — a naïve approximation which holds well in three quadrants of the plane, but overestimates the probability when both $X$ and $Y$ are unlikely. In this quadrant, when both $X$ and $Y$ are both unlikely, a better approximation for $\mathrm{AND_{IL}}$ is the sum of their logits.

We thus propose $\mathrm{AND_{AIL}}$, a linear-approximate AND function for independent logits (AIL, i.e. approximate IL).

$$\mathrm{AND_{AIL}}(x, y) := \begin{cases} x + y, & x < 0, \, y < 0 \\ \min(x, y), & \text{otherwise} \end{cases} \tag{2}$$

As shown in Figure 1 (left, middle), we observe that their output values and shape are very similar.

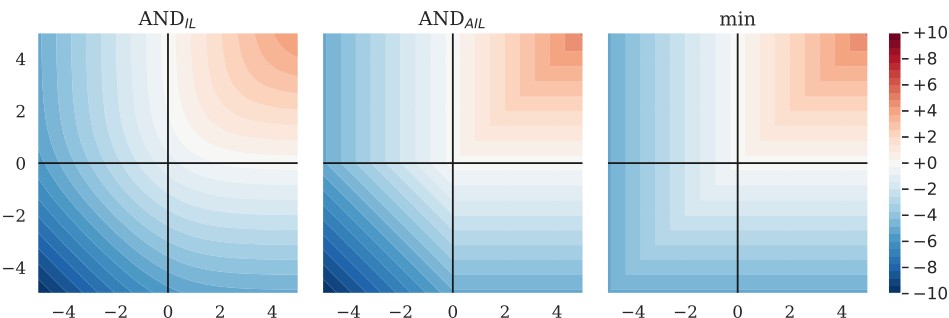

Figure 1: Heatmap comparing the outputs for the exact logit-space probabilistic-and function for independent logits, $\mathrm{AND_{IL}}(x, y)$; our constructed approximation, $\mathrm{AND_{AIL}}(x, y)$; and $\max(x, y)$.

## 2.2 OR

Similarly, we can construct the logit-space OR function, for independent logits. For a pair of logits $x$ and $y$, the probability that either of the corresponding events is true is given by $p = 1 - \sigma(-x)\,\sigma(-y)$. This can be converted into a logit as

$$\mathrm{OR}_{\mathrm{IL}}(x, y) := \mathrm{logit}(\mathrm{P}(X \vee Y)_{x \perp\!\!\!\perp y}) = \log\left(\frac{p}{1-p}\right), \text{ where } p = 1 - \sigma(-x)\,\sigma(-y) \quad (3)$$

which can be roughly approximated by the max function. This is equivalent to setting the probability of **either** of event $X$ or $Y$ occurring to be equal to the probability of the **most likely** event. This underestimates the upper-right quadrant (below), which we can approximate better as the sum of the input logits, yielding

$$\mathrm{OR}_{\mathrm{AIL}}(x, y) := \begin{cases} x + y, & x > 0,\ y > 0 \\ \max(x, y), & \text{otherwise} \end{cases} \quad (4)$$

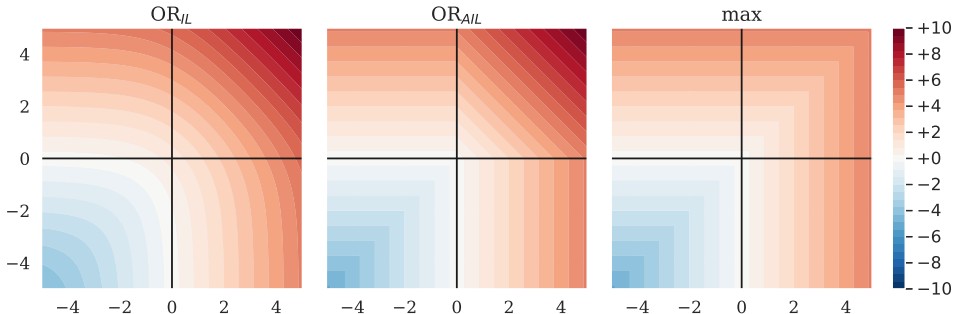

Figure 2: Comparison of the exact logit-space probabilistic-or function for independent logits, $\mathrm{OR}_{\mathrm{IL}}(x, y)$; our constructed approximation, $\mathrm{OR}_{\mathrm{AIL}}(x, y)$; and $\max(x, y)$.

## 2.3 XNOR

We also consider the construction of a logit-space XNOR operator. This is the probability that $X$ and $Y$ occur either together, or not at all, given by

$$\mathrm{XNOR}_{\mathrm{IL}}(x, y) := \mathrm{logit}(\mathrm{P}(X \bar\oplus Y)_{x \perp\!\!\!\perp y}) = \log\left(\frac{p}{1-p}\right), \quad (5)$$

where $p = \sigma(x)\,\sigma(y) + \sigma(-x)\,\sigma(-y)$. We can approximate this with

$$\mathrm{XNOR}_{\mathrm{AIL}}(x, y) := \mathrm{sgn}(xy)\min(|x|, |y|), \quad (6)$$

which focuses on the logit of the feature **most likely** to **flip** the expected **parity** (Figure 3).

We could use other approximations, such as the sign-preserving geometric mean,

$$\mathrm{signed\_geomean}(x, y) := \mathrm{sgn}(xy)\sqrt{|xy|}, \quad (7)$$

but note that the gradient of this is divergent, both along $x = 0$ and along $y = 0$.

## 2.4 DISCUSSION

By working via probabilities, and assuming inputs encode independent events, we have derived logit-space equivalents of the boolean logic functions, AND, OR, and XNOR. Since these are computationally demanding to compute repeatedly within a neural network, we have constructed approximations of them: $\mathrm{AND}_{\mathrm{AIL}}$, $\mathrm{OR}_{\mathrm{AIL}}$, and $\mathrm{XNOR}_{\mathrm{AIL}}$. Like ReLU, these involve only comparison, addition, and multiplication operations which are cheap to perform. In fact, $\mathrm{AND}_{\mathrm{AIL}}$ and $\mathrm{OR}_{\mathrm{AIL}}$ are a generalization of ReLU to an extra dimension, since $\mathrm{OR}_{\mathrm{AIL}}(x, y = 0) = max(x, 0)$.

The majority of activation functions are one-dimensional, $f : \mathbb{R} \to \mathbb{R}$. In contrast to this, our proposed activation functions are all two-dimensional, $f : \mathbb{R}^2 \to \mathbb{R}$. They must be applied to pairs of features

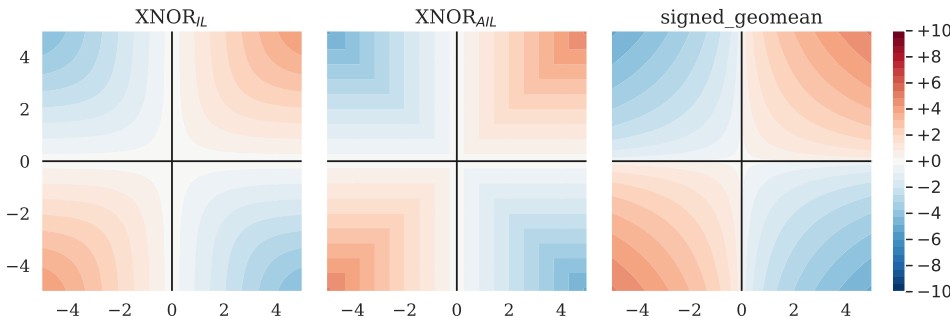

Figure 3: Comparison of the exact logit-space probabilistic-xnor function for independent logits, $\text{XNOR}_{\text{IL}}(x, y)$; our constructed approximation, $\text{XNOR}_{\text{AIL}}(x, y)$; and $\text{signed\_geomean}(x, y)$.

from the embedding space, and will reduce the dimensionality of the input space by a factor of 2. This behaviour is the same as seen in MaxOut networks (Goodfellow et al., 2013) which use $\max$ as their activation function, $\text{MaxOut}(x, y; k) := \max(x, y)$. Similar to MaxOut, our activation functions could be generalised to higher dimensional inputs, $f : \mathbb{R}^k \to \mathbb{R}$, by considering the behaviour of the logit-space AND, OR, XNOR operations with regard to more inputs. For simplicity, we restrict this work to consider only $k{=}2$, but note these activation functions also generalize to higher dimensions.

## 2.5 ENSEMBLING

By using multiple logit-boolean activation functions **simultaneously** alongside each other, we permit the network multiple options of how to relate features together. When combining the activation functions, we considered two strategies.

In the partition (p) strategy, we split the $n_c$ dimensional pre-activation embedding equally into $m$ partitions, apply different activation functions on each partition, and concatenate the results together. Using AIL activation functions under this strategy, the output dimension will always be half that of the input, as it is for each AIL activation function individually.

In the duplication (d) strategy, we apply $m$ different activation functions in parallel to the same $n_c$ elements. The output is consequently larger, with dimension $m\,n_c$. If desired, we can counteract the $2{\to}1$ reduction of AIL activation functions by using two of them together under this strategy.

Utilising $\text{AND}_{\text{AIL}}$, $\text{OR}_{\text{AIL}}$ and $\text{XNOR}_{\text{AIL}}$ simultaneously allows our networks to access logit-space equivalent of 12 of the 16 boolean logical operations with only a single sign inversion (in either one of the inputs or the output). Including the bias term and skip connections, the network has easy access to logit-space equivalents of all 16 boolean logical operations.

## 3 EXPERIMENTS

We evaluated the performance of our AIL activation functions, both individually and together in an ensemble, on a range of benchmarking tasks. Since $\text{AND}_{\text{AIL}}$ and $\text{OR}_{\text{AIL}}$ are equivalent when the sign of operands and outputs can be freely chosen, we evaluate only on $\text{OR}_{\text{AIL}}$ and not both.

We compared the AIL activation functions against three primary baselines: (1) ReLU, (2) $\max(x, y) = \text{MaxOut}([x, y]; k{=}2)$, and (3) the concatenation of $\max(x, y)$ and $\min(x, y)$, denoted $\{\text{Max}, \text{Min (d)}\}$. The $\{\text{Max}, \text{Min (d)}\}$ ensemble is equivalent to $\text{GroupSort}$ with a group size of 2 (Anil et al., 2019; Chernodub & Nowicki, 2017), sometimes referred to as the $\text{MaxMin}$ operator, and is comparable to the concatenation of $\text{OR}_{\text{AIL}}$ and $\text{AND}_{\text{AIL}}$ under our duplication strategy.

### 3.1 PARITY

In a simple initial experiment, we constructed a synthetic dataset whose labels could be derived directly using the logical operation XNOR. Each sample in this dataset consisted of four input logit values, with a label that was derived by converting each logit to probability space, rounding to the

nearest integer, and taking the parity over this set of binary digits (i.e. true when we have an even number of activated bits, false otherwise).

A very small MLP model with two hidden layers (the first with four neurons, the second with two neurons) should be capable of perfect classification accuracy on this dataset with a sparse weight matrix by learning to extract pairwise binary relationships between inputs using $\text{XNOR}_{\text{AIL}}$. We trained such an MLP, for 100 epochs using Adam, one-cycle learning rate schedule, max LR 0.01, weight decay $1 \times 10^{-4}$.

The two-layer MLP using the $\text{XNOR}_{\text{AIL}}$ activation learned a sparse weight matrix able to perfectly classify any input combination, shown in Figure 4. In comparison, an identical network setup using ReLU was only able to produce 60% classification accuracy. Though this accuracy could be improved by increasing the MLP width or depth, the weight matrix was still not sparse. This experiment provides an example situation where $\text{XNOR}_{\text{AIL}}$ is utilized by a network to directly extract information about the relationships between network inputs. For additional results, see Appendix A.7.

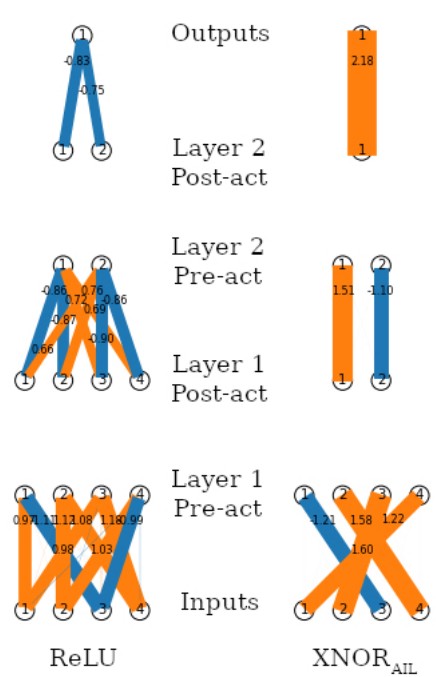

Figure 4: Visualisation of weight matrices learnt by two-layer MLPs on a binary classification task, where the target output is the parity of the inputs. Widths of lines indicate weight magnitudes (orange: +ve values, blue: -ve values). Network with ReLU: 60% accuracy. Network with $\text{XNOR}_{\text{AIL}}$: 100% accuracy.

### 3.2 MLP ON COVERTYPE

We trained small one, two, and three-layer MLP models on the Covertype dataset (UCI Machine Learning Repository, 1998; Blackard & Dean, 1998) from the UCI Machine Learning Repository. The Covertype dataset is a classification task consisting of $581\,012$ samples of forest coverage across 7 classes. Each sample has 54 attributes. We used a fixed random 80:20 train:test split; for training details and accuracy vs model size, see Appendix A.8.

For each activation function, we varied the number of hidden units per layer to investigate how the activation functions affected the performance of the networks as its capacity changed. We did not use weight decay or data augmentation for this experiment, and so the network exhibits clear overfitting with larger architectures. As shown in Figure 16, Appendix A.8, $\text{XNOR}_{\text{AIL}}$ performs significantly best on Covertype ($p < 10^{-5}$) for MLP with 2 or 3 layers, followed by networks which include $\text{XNOR}_{\text{AIL}}$ alongside with other activations, and $\text{signed\_geomean}$. The duplication strategy outperforms partition.

### 3.3 MLP ON BACH CHORALES AND LOGIT INDEPENDENCE

The Bach Chorale dataset (Boulanger-Lewandowski et al., 2012) consists of 382 chorales composed by JS Bach, each ~12 measures long, totalling approximately 83,000 notes. Represented as discrete sequences of tokens, it has served as a benchmark for music processing for decades, from heuristic methods to HMMs, RNNs, and CNNs (Mozer, 1990; Hild et al., 1992; Allan & Williams, 2005; Liang, 2016; Hadjeres et al., 2017; Huang et al., 2019). The chorales are comprised of 4 voices (melodic lines) whose behaviour is guided by soft musical rules that depend on the prior movement of that voice as well as the movement of the other voices. An example of one such rule is that two voices a fifth apart ought not to move in parallel with one another. The task we choose here is to determine whether a given short four-part musical excerpt is taken from a Bach chorale or not. During training, we stochastically corrupt chorale excerpts to provide negative examples (see Appendix A.9). We

trained 2-layer MLP discriminators with a single sigmoid output and exchanged all other activation functions (summarized in Figure 17). We found that {OR, AND, XNOR$_{\text{AIL}}$ (d)} performed best, but that overall the results were comparable ($p < 0.1$ between best and worst, Student's t-test between 10 random initialisations).

Additionally, we investigated the independence between logits in the trained pre-activation embeddings. We would expect that an MLP which is optimally engaging its neurons would maintain independence between features in order to maximize information. To capture the existence of correlations, we took the cosine similarity between rows of the weight matrix. Since the inputs to all features in a given layer are the same, this is equivalent to measuring the similarity between corresponding pair of pre-activation features. We performed two experiments. In the first, we measured correlations between all pairwise combinations, and in the second we took correlations between only adjacent pre-activations that would be paired for the logical activation functions. For these experiments we used 2 hidden layers and a width that showed maximal performance for each activation function. The results are shown in Appendix A.10.

## 3.4 CNN AND MLP ON MNIST

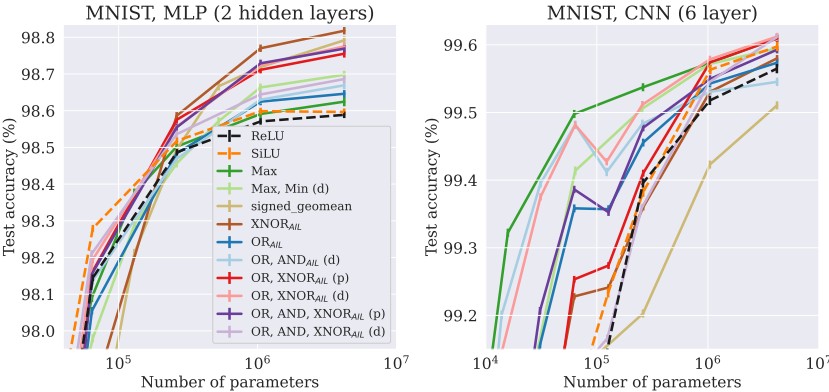

Figure 5: We trained CNN on MNIST, MLP on flattened-MNIST, using Adam (1-cycle, 10 ep), hyperparams determined by random search. Mean (bars: std dev) of $n = 40$ weight inits.

We trained 2-layer MLP and 6-layer CNN models on MNIST with Adam (Kingma & Ba, 2017), 1-cycle schedule (Smith & Topin, 2017; Smith, 2018), and using hyperparameters tuned through a random search against a validation set comprised of the last 10k images of the training partition.

The MLP used two hidden layers, the widths of which were varied together to evaluate the performance for a range of model sizes. The CNN used six layers of 3x3 convolution layers, with 2x2 max pooling (stride 2) after every other conv layer. The output was flattened and followed by three MLP layers. The widths of the layers were scaled up approximately linearly to explore a range of model sizes (see Appendix A.11 for more details).

For the MLP, XNOR$_{\text{AIL}}$ performed best along with signed_geomean ($p < 0.1$), ahead of all other activations ($p < 0.01$; Figure 5 left panel). With the CNN, five activation configurations ({OR, AND, XNOR$_{\text{AIL}}$ (p)}, {OR, XNOR$_{\text{AIL}}$ (d/p)}, Max, and SiLU) performed best ($p < 0.05$; Figure 5, right panel). Additionally, we note that CNNs which used OR$_{\text{AIL}}$ or Max (alone or in an ensemble) maintained high performance with an order of magnitude fewer parameters ($3 \times 10^4$) than networks which did not ($3 \times 10^5$ params).

## 3.5 RESNET50 ON CIFAR-10/100

We explored the impact of our AIL activation functions on the performance of deep networks by deploying them within a pre-activation ResNet50 model (He et al., 2015; 2016). We exchanged all ReLU activation functions in the network to a candidate activation function while maintaining the size of pass-through embedding. We experimented with changing the width of the network, scaling up the embedding space and all hidden layers by a common factor. The network was trained on

CIFAR-10/-100 for 100 epochs using Adam (Kingma & Ba, 2017), 1-cycle (Smith, 2018; Smith & Topin, 2017). Hyperparameters were optimized through a random search against a validation partition of CIFAR-100 for a fixed width factor $w = 2$ only. The same hyperparameters were reused for experiments of changing width, and for CIFAR-10. We used width factors of 0.5, 1, 2, and 4 for $1 \rightarrow 1$ activation functions (ReLU, etc), widths of 0.75, 1.5, 3, and 6 for $2 \rightarrow 1$ activation functions (Max, etc), and widths of 0.4, 0.75, 1.5, and 3 for $\{OR_{AIL}, AND_{AIL}, XNOR_{AIL}$ (d)$\}$.

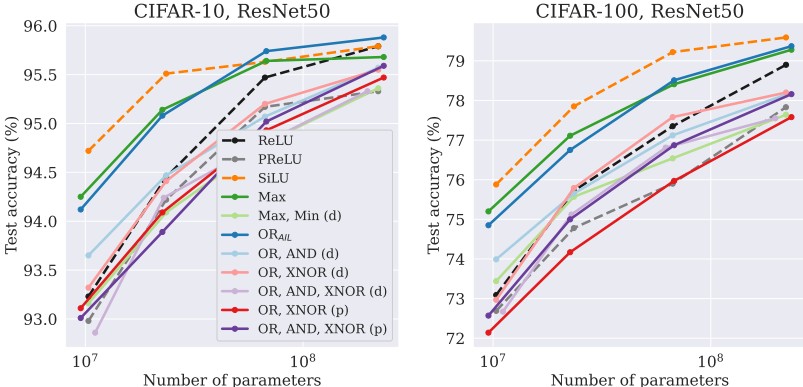

Figure 6: ResNet50 on CIFAR-10/100, varying the activation function. The same activation function (or ensemble) was used through the network. The width was varied to explore a range of network sizes (see text). Trained for 100 ep. with Adam, using hyperparams determined by random search on CIFAR-100, $w = 2$, only.

For both CIFAR-10 and 100, we find that SiLU, $OR_{AIL}$, and Max outperform ReLU across a wide range of width values, as shown in Figure 6. These three activation functions hold up their performance best as the number of parameters is reduced. Note that SiLU was previously discovered as an optimal activation function for deep residual image models (Ramachandran et al., 2017), and so it is expected to perform highly in this setting. Meanwhile, other AIL activation functions perform similarly to ReLU when the width is thin, and slightly worse than ReLU when the width is wide. When used on its own and not part of an ensemble, the $XNOR_{AIL}$ activation function performed very poorly (off the bottom of the chart), indicating it is not well suited for this task.

## 3.6 TRANSFER LEARNING

We considered the task of transfer learning on several image classification datasets. We used a ResNet18 model (He et al., 2015) pretrained on ImageNet-1k. The weights were frozen (not fine-tuned) and used to generate embeddings of samples from other image datasets. We trained a two-layer MLP to classify the images from these embeddings, using various activation functions. For a comprehensive set of baselines, we compared against every activation function built into PyTorch 1.10. To make the number of parameters similar, we used a width of 512 for activation functions with $1 \rightarrow 1$ mapping (e.g. ReLU), a width of 650 for activation functions with a $2 \rightarrow 1$ mapping (e.g. Max, $OR_{AIL}$), and a width of 438 for $\{OR, AND, XNOR_{AIL}$ (d)$\}$. See Appendix A.13 for more details.

Our results are shown in Table 1. We found that all our activation functions outperformed ReLU on every transfer task. In particular, our three activation ensembles using the duplication strategy performed very highly across all 7 transfer learning tasks — overall, $\{OR, AND, XNOR_{AIL}$ (d)$\}$ appears to perform best (top for 3 datasets, second on 3), followed by $\{OR, AND_{AIL}$ (d)$\}$. Our activation functions were beaten on Caltech101, Oxford Flowers, and Stanford Cars, and only by PReLU. For these three datasets, a linear readout outperformed an MLP with ReLU, and on Caltech101 the linear layer performed best. This suggests that PReLU excels only on particularly simple tasks. For further discussion, see Appendix A.13.

## 3.7 ADDITIONAL RESULTS

For results on abstract reasoning and compositional zero-shot learning, please see Appendix A.14 and Appendix A.15, respectively.

Table 1: Transfer learning from a frozen ResNet-18 architecture pretrained on ImageNet-1k to other computer vision datasets. An MLP head with two layers of pre-activation width of either 438, 512, or 650 (depending on activation function, to keep the number of params approximately constant) was trained, without re-training the pretrained base model. Trained with SGD, cosine annealed LR 0.01, for 25 epochs. Mean (standard error) of $n = 5$ random initializations of the MLP (same pretrained network). Bold: best. Underlined: second best. Italic: no significant difference from best (two-sided Student's t-test, $p > 0.05$). Background: linear color scale from ReLU baseline (white) to best (black).

| | Test Accuracy (%) | | | | | | |
| Activation function | Caltech101 | CIFAR10 | CIFAR100 | Flowers | Cars | STL-10 | SVHN |
|---|---|---|---|---|---|---|---|
| Linear layer only | $88.35_{\pm 0.15}$ | $78.56_{\pm 0.09}$ | $57.39_{\pm 0.09}$ | $92.32_{\pm 0.20}$ | $33.51_{\pm 0.06}$ | $94.68_{\pm 0.02}$ | $45.42_{\pm 0.06}$ |
| ReLU | $86.58_{\pm 0.17}$ | $81.63_{\pm 0.05}$ | $58.04_{\pm 0.11}$ | $90.71_{\pm 0.26}$ | $30.97_{\pm 0.26}$ | $94.62_{\pm 0.06}$ | $51.39_{\pm 0.06}$ |
| LeakyReLU | $86.60_{\pm 0.13}$ | $81.67_{\pm 0.11}$ | $58.01_{\pm 0.09}$ | $90.73_{\pm 0.32}$ | $31.09_{\pm 0.24}$ | $94.61_{\pm 0.05}$ | $51.40_{\pm 0.05}$ |
| PReLU | $87.83_{\pm 0.21}$ | $81.03_{\pm 0.13}$ | $58.90_{\pm 0.18}$ | $93.17_{\pm 0.19}$ | $39.84_{\pm 0.18}$ | $94.54_{\pm 0.05}$ | $51.42_{\pm 0.09}$ |
| Softplus | $86.16_{\pm 0.18}$ | $79.13_{\pm 0.08}$ | $56.58_{\pm 0.07}$ | $89.39_{\pm 0.29}$ | $21.23_{\pm 0.13}$ | $94.63_{\pm 0.03}$ | $47.44_{\pm 0.06}$ |
| ELU | $87.18_{\pm 0.09}$ | $80.44_{\pm 0.08}$ | $58.08_{\pm 0.10}$ | $91.71_{\pm 0.14}$ | $34.70_{\pm 0.06}$ | $94.55_{\pm 0.05}$ | $50.07_{\pm 0.07}$ |
| CELU | $87.18_{\pm 0.09}$ | $80.44_{\pm 0.08}$ | $58.08_{\pm 0.10}$ | $91.71_{\pm 0.14}$ | $34.70_{\pm 0.06}$ | $94.55_{\pm 0.05}$ | $50.07_{\pm 0.07}$ |
| SELU | $87.74_{\pm 0.09}$ | $79.93_{\pm 0.13}$ | $58.24_{\pm 0.06}$ | $92.27_{\pm 0.13}$ | $37.51_{\pm 0.17}$ | $94.53_{\pm 0.07}$ | $49.38_{\pm 0.06}$ |
| GELU | $87.10_{\pm 0.15}$ | $81.39_{\pm 0.09}$ | $58.51_{\pm 0.13}$ | $91.51_{\pm 0.15}$ | $33.43_{\pm 0.15}$ | $94.62_{\pm 0.06}$ | $51.56_{\pm 0.08}$ |
| SiLU | $86.91_{\pm 0.11}$ | $80.53_{\pm 0.11}$ | $58.14_{\pm 0.12}$ | $91.37_{\pm 0.18}$ | $32.15_{\pm 0.17}$ | $94.59_{\pm 0.05}$ | $50.69_{\pm 0.06}$ |
| Hardswish | $87.12_{\pm 0.12}$ | $80.10_{\pm 0.10}$ | $58.25_{\pm 0.10}$ | $91.56_{\pm 0.25}$ | $33.17_{\pm 0.23}$ | $94.62_{\pm 0.05}$ | $50.09_{\pm 0.09}$ |
| Mish | $87.11_{\pm 0.12}$ | $81.09_{\pm 0.11}$ | $58.37_{\pm 0.10}$ | $91.61_{\pm 0.15}$ | $33.75_{\pm 0.14}$ | $94.61_{\pm 0.05}$ | $51.29_{\pm 0.08}$ |
| Softsign | $81.47_{\pm 0.18}$ | $80.03_{\pm 0.09}$ | $54.84_{\pm 0.09}$ | $82.34_{\pm 0.22}$ | $17.33_{\pm 0.10}$ | $94.70_{\pm 0.03}$ | $49.48_{\pm 0.07}$ |
| Tanh | $87.48_{\pm 0.06}$ | $80.56_{\pm 0.07}$ | $57.35_{\pm 0.08}$ | $90.32_{\pm 0.20}$ | $29.51_{\pm 0.12}$ | $94.63_{\pm 0.07}$ | $50.15_{\pm 0.08}$ |
| GLU | $86.71_{\pm 0.31}$ | $79.19_{\pm 0.07}$ | $57.64_{\pm 0.10}$ | $90.34_{\pm 0.19}$ | $27.04_{\pm 0.12}$ | $94.57_{\pm 0.03}$ | $48.28_{\pm 0.17}$ |
| Max | $86.96_{\pm 0.20}$ | $81.76_{\pm 0.14}$ | $58.60_{\pm 0.12}$ | $90.98_{\pm 0.18}$ | $33.37_{\pm 0.15}$ | $94.70_{\pm 0.06}$ | $51.36_{\pm 0.12}$ |
| Max, Min (d) | $87.23_{\pm 0.13}$ | $82.31_{\pm 0.10}$ | $59.05_{\pm 0.10}$ | $91.68_{\pm 0.18}$ | $34.91_{\pm 0.12}$ | $94.64_{\pm 0.04}$ | $51.72_{\pm 0.04}$ |
| $\text{XNOR}_{\text{AIL}}$ | $86.97_{\pm 0.18}$ | $81.83_{\pm 0.06}$ | $58.46_{\pm 0.10}$ | $90.93_{\pm 0.15}$ | $32.56_{\pm 0.10}$ | $94.71_{\pm 0.06}$ | $51.54_{\pm 0.05}$ |
| $\text{OR}_{\text{AIL}}$ | $87.45_{\pm 0.14}$ | $81.88_{\pm 0.07}$ | $59.10_{\pm 0.09}$ | $92.00_{\pm 0.15}$ | $36.01_{\pm 0.12}$ | $94.69_{\pm 0.04}$ | $51.52_{\pm 0.07}$ |
| OR, $\text{AND}_{\text{AIL}}$ (d) | $87.43_{\pm 0.11}$ | $82.38_{\pm 0.06}$ | $59.90_{\pm 0.08}$ | $92.07_{\pm 0.18}$ | $37.16_{\pm 0.15}$ | $94.55_{\pm 0.05}$ | $52.11_{\pm 0.09}$ |
| OR, $\text{XNOR}_{\text{AIL}}$ (p) | $87.42_{\pm 0.12}$ | $81.92_{\pm 0.07}$ | $59.09_{\pm 0.10}$ | $91.93_{\pm 0.12}$ | $35.99_{\pm 0.17}$ | $94.68_{\pm 0.03}$ | $51.43_{\pm 0.08}$ |
| OR, $\text{XNOR}_{\text{AIL}}$ (d) | $87.09_{\pm 0.21}$ | $82.20_{\pm 0.04}$ | $59.44_{\pm 0.07}$ | $91.90_{\pm 0.10}$ | $36.88_{\pm 0.10}$ | $94.69_{\pm 0.06}$ | $52.02_{\pm 0.16}$ |
| OR, AND, $\text{XNOR}_{\text{AIL}}$ (p) | $87.43_{\pm 0.14}$ | $81.78_{\pm 0.06}$ | $59.27_{\pm 0.13}$ | $91.98_{\pm 0.29}$ | $35.90_{\pm 0.09}$ | $94.66_{\pm 0.03}$ | $51.47_{\pm 0.13}$ |
| OR, AND, $\text{XNOR}_{\text{AIL}}$ (d) | $87.49_{\pm 0.11}$ | $82.50_{\pm 0.08}$ | $59.83_{\pm 0.12}$ | $92.37_{\pm 0.08}$ | $37.60_{\pm 0.20}$ | $94.72_{\pm 0.02}$ | $52.23_{\pm 0.14}$ |

## 4 DISCUSSION

In this work we motivated and introduced novel activation functions analogous to boolean operators in logit-space. We designed the AIL functions, fast approximates to the true logit-space functions equivalent to manipulating the corresponding probabilities, and demonstrated their effectiveness on a wide range of tasks.

Although our activation functions assume independence (which is generally approximately true for the pre-activation features learnt with 1d activation functions), we found the network learnt to induce anti-correlations between features which were paired together by our activation functions (Appendix A.10). This suggests that the assumption of independence is not essential to the performance of our proposed activation functions.

We found that the $\text{XNOR}_{\text{AIL}}$ activation function was highly effective in the setting of shallow networks. Meanwhile, the $\text{OR}_{\text{AIL}}$ activation function was highly effective for representation learning in the setting of a deep ResNet architecture trained on images. In scenarios which involve manipulating high-level features extracted by an embedding network, we find that using an ensemble of AIL activation functions together works best, and that the duplication ensembling strategy outperforms partitioning. In this work we have restricted ourselves to only considering using a single activation function (or ensemble) throughout the network, however our results together indicate that stronger results may be found by using $\text{OR}_{\text{AIL}}$ for feature extraction and an ensemble of $\{\text{OR}, \text{AND}, \text{XNOR}_{\text{AIL}}$ (d)$\}$ for later higher-order reasoning layers within the network.

The idea we propose is nascent and there is a great deal of scope for exploring other forms of activation functions that combine multiple pre-activation features by utilizing "higher-order activation functions".

REPRODUCIBILITY STATEMENT

We have shared code to run all experiments considered in this paper with the reviewers via an anonymous download URL. The code base contains detailed instructions on how to setup each experiment, including downloading the datasets and installing environments with pinned dependencies. It should be possible to reproduce all our experimental results with this code. For the final version of the paper, we will make the code publicly available on an online repository and share a link to it within the paper.

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

# A APPENDIX

## A.1 LINEAR RELU EXAMPLES

In Figure 7, we show a representation of what a 2-d linear layer followed by the ReLU activation function looks like. The output is the same up to rotation.

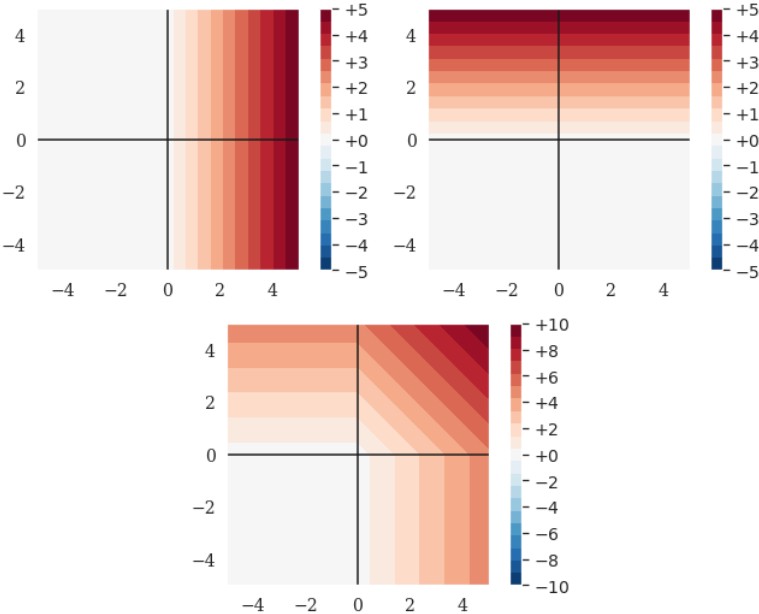

Figure 7: ReLU unit, and ReLU units followed by a linear layer to try to approximate $\text{OR}_{\text{IL}}$, leaving a dead space where the negative logits should be.

If we apply a second linear layer on top of the outputs of the first two units, we can try to approximate the logit AND or OR function. However, the solution using ReLU leaves a quadrant of the output space hollowed out as zero due to its behaviour at truncating away information.

## A.2 SOLVING XOR

A long-standing criticism of artificial neural networks is their inability to solve XOR with a single layer (Minsky & Papert, 1969). Of course, adding a single hidden layer allows a network using ReLU to solve XOR. However, the way that it solves the problem is to join two of the disconnected regions together in a stripe (see Figure 8). Meanwhile, our $\text{XNOR}_{\text{AIL}}$ is trivial able to solve the XOR problem without any hidden layers. For comparison here, we include one hidden layer with 2 units for each network. Including a layer before the activation function makes the task harder for $\text{XNOR}_{\text{AIL}}$, which must learn how to project the input space in order to compute the desired separation. Also, including the linear layer allows the network to generalise to rotations and offset versions of the task.

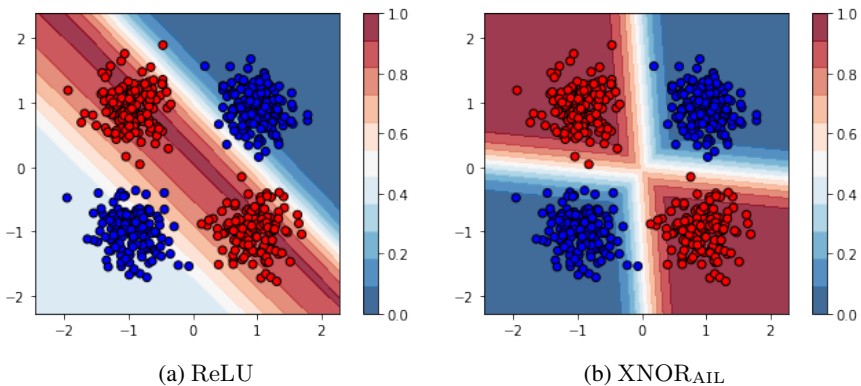

$$\text{(a) ReLU} \qquad\qquad\qquad\qquad\qquad \text{(b) XNOR}_{\text{AIL}}$$

Figure 8: Solving XOR with a single hidden layer of 2 units, using either ReLU or $\text{XNOR}_{\text{AIL}}$ activation. Circles indicate negative (blue) and positive (red) training samples. The heatmaps indicate the output probabilities of the two networks.

### A.3  Linear layers and Bayes' Rule in log-odds form

Bayes' Theorem or Bayes' Rule is given by

$$\mathrm{P}(H|X) = \frac{\mathrm{P}(X|H)\,\mathrm{P}(H)}{\mathrm{P}(X)}. \tag{8}$$

In this case, we update the probability of our hypothesis, $H$, based on the event of the observation of a new piece of evidence, $X$. Our prior belief for the hypothesis is $\mathrm{P}(H)$, and posterior is $\mathrm{P}(H|X)$. To update our belief from the prior and yield the posterior, we multiply by the Bayes factor for the evidence which is given by $\mathrm{P}(X|H)/\mathrm{P}(X)$.

Converting the probabilities into log-odds ratios (logits) yields the following representation of Bayes' Rule.

$$\log\left(\frac{\mathrm{P}(H|X)}{\mathrm{P}(H^C|X)}\right) = \log\left(\frac{\mathrm{P}(H)}{\mathrm{P}(H^C)}\right) + \log\left(\frac{\mathrm{P}(X|H)}{\mathrm{P}(X|H^C)}\right) \tag{9}$$

Here, $H^C$ is the complement to $H$ (the event that the hypothesis is false), and $\mathrm{P}(H^C) = 1 - \mathrm{P}(H)$. Our prior log-odds ratio is $\log\left(\mathrm{P}(H)/\mathrm{P}(H^C)\right)$, and our posterior after updating based on the observation of new evidence $X$ is $\log\left(\mathrm{P}(H|X)/\mathrm{P}(H^C|X)\right)$. To update our belief from the prior and yield the posterior, we add the log-odds Bayes factor for the evidence which is given by $\log\left(\mathrm{P}(X|H)/\mathrm{P}(X|H^C)\right)$.

In log-odds space, a series of updates with multiple pieces of independent evidence can be performed at once with a summation operation.

$$\log\left(\frac{\mathrm{P}(H|\boldsymbol{x})}{\mathrm{P}(H^C|\boldsymbol{x})}\right) = \log\left(\frac{\mathrm{P}(H)}{\mathrm{P}(H^C)}\right) + \sum_i \log\left(\frac{\mathrm{P}(X_i|H)}{\mathrm{P}(X_i|H^C)}\right). \tag{10}$$

This is operation can be represented by the linear layer in an artificial neural network, $z_k = b_k + \boldsymbol{w}_k^T \boldsymbol{x}$. Here, the bias term $b_k = \log\left(\mathrm{P}(H)/\mathrm{P}(H^C)\right)$ is the prior for hypothesis (the presence of the feature represented by the k-th neuron), and the series of weighted inputs from the previous layer, $w_{ki}\,x_i$ provide evidential updates. This is also equivalent to the operation of a multinomial naïve Bayes classifier, expressed in log-space, if we choose $w_{ki} = \log p_{ki}$ (Rennie et al., 2003).

### A.4  Difference between AIL and IL functions

Here, we measure and show the difference between the true logit-space operations and our AIL approximations, shown in Figure 9, Figure 10, and Figure 11.

In each case, we observe that the magnitude of the difference is never more than 1, which occurs along the boundary lines in AIL. Since the magnitude of the three functions increase as we move away from the origin, the relative difference decreases in magnitude as the size of $x$ and $y$ increase.

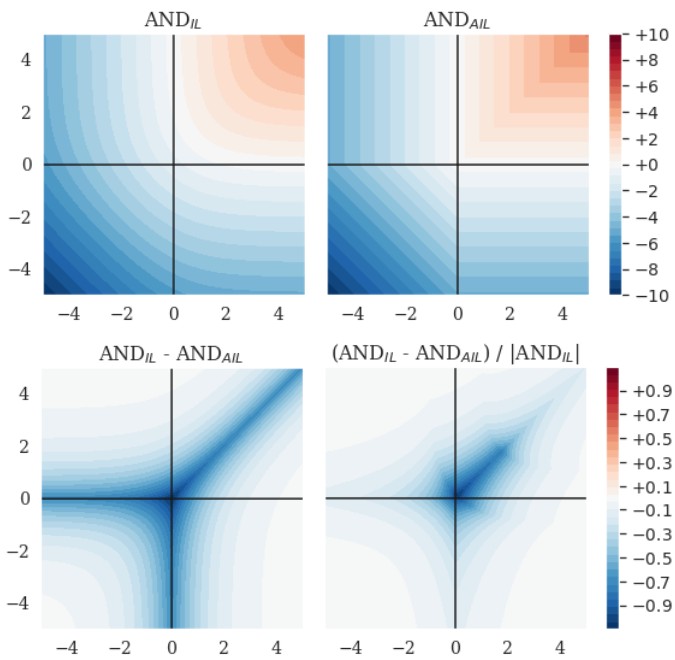

Figure 9: Heatmaps showing $\text{AND}_{\text{IL}}$, $\text{AND}_{\text{AIL}}$, their difference, and their relative difference.

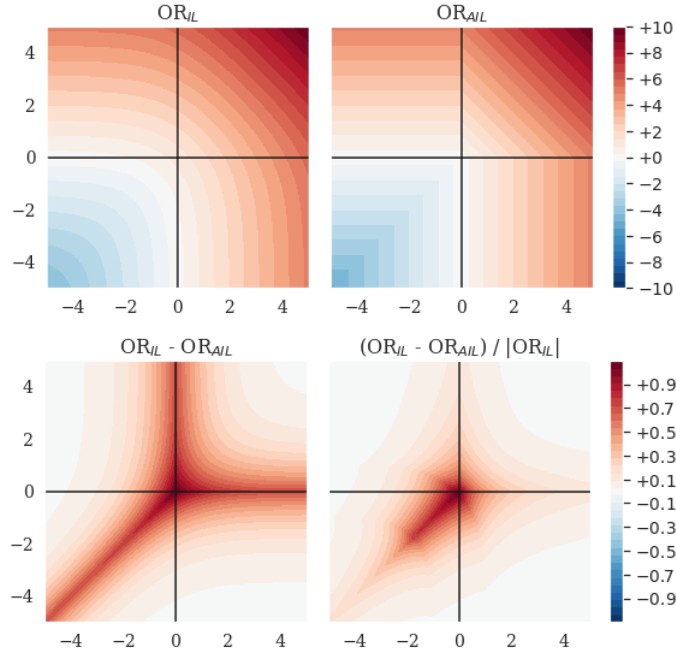

Figure 10: Heatmaps showing $\text{OR}_{\text{IL}}$, $\text{OR}_{\text{AIL}}$, their difference, and their relative difference.

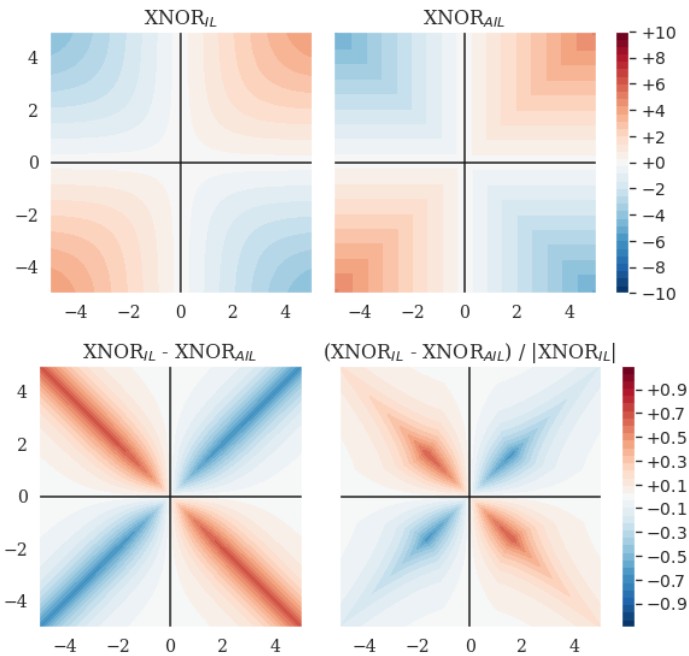

Figure 11: Heatmaps showing $\text{XNOR}_{\text{IL}}$, $\text{XNOR}_{\text{AIL}}$, their difference, and their relative difference.

## A.5 Gradient of AIL and IL functions

We show the gradient of each of the logit-space boolean operators and their AIL approximates in Figure 12, Figure 13, and Figure 14. By the symmetry of each of the functions, the derivative with respect to $y$ is a reflected copy of the gradient with respect to $x$.

We find that the gradient of each AIL function closely matches that of the exact form. Whilst there are "dead" regions where the gradient is zero, this only occurs for one of the derivatives at a time (there is always a gradient with respect to at least one of $x$ and $y$).

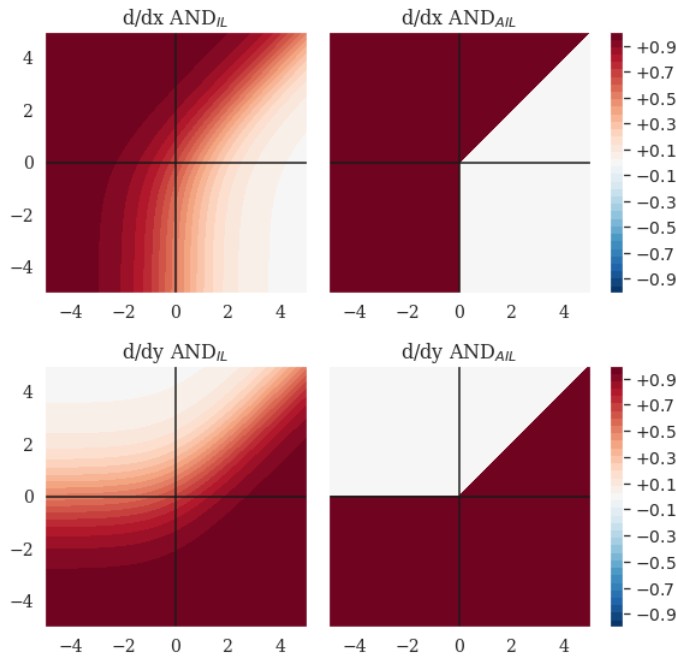

Figure 12: Heatmaps showing the gradient with respect to $x$ and $y$ of $\mathrm{AND_{IL}}$ and $\mathrm{AND_{AIL}}$.

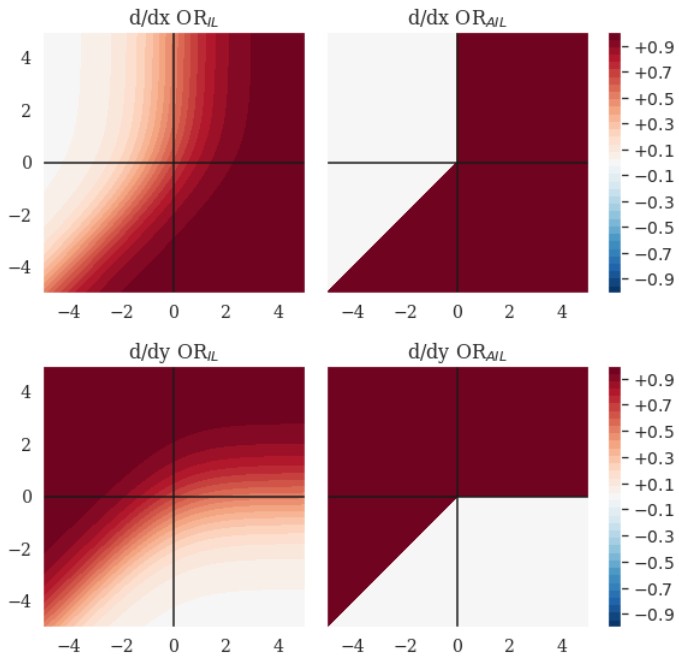

Figure 13: Heatmaps showing the gradient with respect to $x$ and $y$ of $\text{OR}_{\text{IL}}$ or $\text{OR}_{\text{AIL}}$.

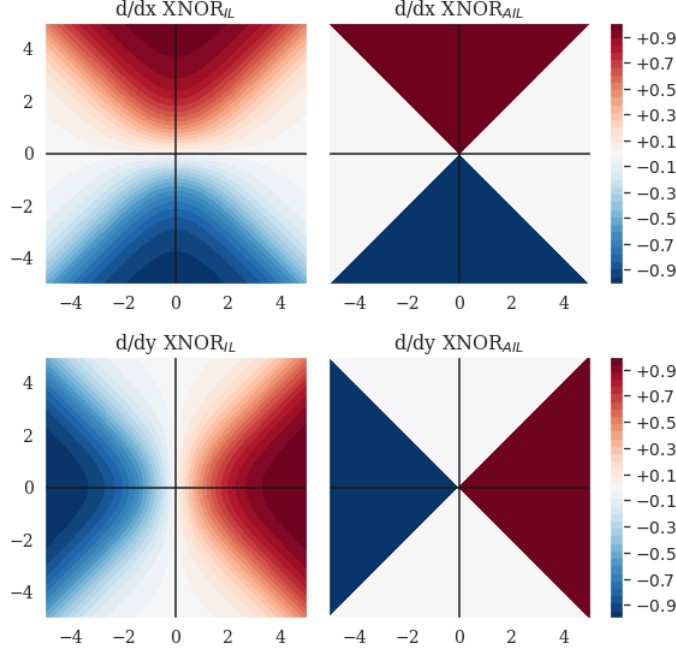

Figure 14: Heatmaps showing the gradient with respect to $x$ and $y$ of $\text{XNOR}_{\text{IL}}$ xnor $\text{XNOR}_{\text{AIL}}$.

## A.6 DATASET SUMMARY

The datasets used in this work are summarised in Table 2.

We used the same splits for Caltech101 as used in Kabir et al. (2020).

Table 2: Dataset summaries.

| Dataset | № Samples | | Classes | Reference |
| | Train | Test | | |
| --- | --- | --- | --- | --- |
| Bach Chorales | 229 | 77 | 2 | Boulanger-Lewandowski et al. (2012) |
| Caltech101 | 6162 | 1695 | 101 | Fei-Fei et al. (2006) |
| CIFAR-10 | 50 000 | 10 000 | 10 | Krizhevsky (2009) |
| CIFAR-100 | 50 000 | 10 000 | 100 | Krizhevsky (2009) |
| Covertype | 464 810 | 116 202 | 7 | Blackard (1998); Blackard & Dean (1998) |
| I-RAVEN | 6000 | 2000 | — | Hu et al. (2020) |
| MIT-States | 30 338 | 12 995 | 245 obj, 115 attr | Isola et al. (2015) |
| MNIST | 60 000 | 10 000 | 10 | LeCun et al. (1998) |
| Oxford Flowers | 6552 | 818 | 102 | Nilsback & Zisserman (2008) |
| Stanford Cars | 8144 | 8041 | 196 | Krause et al. (2013) |
| STL-10 | 5000 | 8000 | 10 | Coates et al. (2011) |
| SVHN | 73 257 | 26 032 | 10 | Netzer et al. (2011) |

The Covertype dataset was chosen as a dataset that contains only simple features (and not pixels of an image) on which we could study a simple MLP architecture, and was selected based on its popularity on the UCI ML repository.

The Bach Chorales dataset was chosen because — in addition to being in continued use by ML researchers for decades — it presents an interesting opportunity to consider a task where logical activation functions are intuitively applicable, as it is a relatively small dataset that has also been approached with rule-based frameworks, e.g. the expert system by Ebcioglu (1988).

MNIST, CIFAR-10, CIFAR-100 are standard image datasets, commonly used. We used small MLP and CNN architectures for the experiments on MNIST so we could investigate the performance of the network for many configurations (varying the size of the network). We used ResNet-50, a very common deep convolutional architecture within the computer vision field, on CIFAR-10/100 to evaluate the performance in the context of a deep network.

The datasets used for the transfer learning task are all common and popular natural image datasets, with some containing coarse-grained classification (CIFAR-10), others fine-grained (Stanford Cars), and with a varying dataset size (5000—75000 training samples). We chose to do an experiment involving transfer learning because it is a common practical situation where one must train only a small network that handles high-level features, and is the sort of situation which involves manipulating high-level features, relying on the pretrained network to do the feature extraction.

We considered other domains where logical reasoning is involved as a component of the task, and isolated abstract reasoning and compositional zero-shot learning as suitable tasks.

For abstract reasoning, we wanted to use an IQ style test, and determined that I-RAVEN was a state-of-the-art dataset within this domain (having fixed some problems with the pre-existing RAVEN dataset). We determined that the SRAN architecture from the paper which introduced I-RAVEN was still state-of-the-art on this task, and so used this.

Another problem domain in which we thought it would be interesting to study our activation functions was compositional zero-shot learning (CZSL). This is because the task inherently involves combining an attribute with an object (i.e. the AND operation). For CZSL, we looked at SOTA methods on https://paperswithcode.com. The best performance was from SymNet, but this was only implemented in TensorFlow and our code was set up in PyTorch so we built our experiments on the back of the second-best instead, which is the TMN architecture. In the TMN paper, they used two datasets: MIT-States and UT-Zappos-1. In our preliminary experiments, we found that the network started overfitting on MIT-States after around 6 epochs, but on UT-Zappos-1 it was overfitting after the first or second epoch (one can not tell beyond the fact the val performance is best for the first epoch). In the context of zero-shot learning, an epoch uses every image once, but there are also only a finite number of tasks in the dataset. Because there are multiple samples for each noun/adjective pair, and each noun only appears with a handful of adjectives and vice versa, there is in a way fewer tasks in one epoch than there are images. Hence it is possible for a zero-shot learning model to overfit to the training tasks in less than one epoch (recall that the network includes a pretrained ResNet model

for extracting features from the images). For simplicity, we dropped UT-Zappos-1 and focused on MIT-States.

## A.7 Parity Experiments

Following on from the parity experiment described in the main text (Section 3.1), we also introduced a second synthetic dataset with a labelling function that, while slightly more complex than the first, was still solvable by applying the logical XNOR operation to the network inputs. In this dataset increased our number of inputs to 8, and derived our labels by applying a set of nested $\text{XNOR}_{\text{IL}}$ operations:

$$\text{XNOR}_{\text{IL}}(\, \text{XNOR}_{\text{IL}}(\text{XNOR}_{\text{IL}}(x_2, x_5), \text{XNOR}_{\text{IL}}(x_3, x_4)),$$
$$\text{XNOR}_{\text{IL}}(\text{XNOR}_{\text{IL}}(x_6, x_7), \text{XNOR}_{\text{IL}}(x_0, x_1)).$$

For this more difficult task we also reformulated our initial experiment into a regression problem, as the continuous targets produced by this labelling function are more informative than the rounded binary targets used in the first experiment. We also adjusted our network setup to have an equal number of neurons at each hidden layer as we found that this significantly improved model performance[2]. We again trained using the same model hyper-parameters for 100 epochs.

While this time the model was not able to learn a sparse weight matrix that exactly reflected our labelling function (see Figure 15), the model was again able to leverage the $\text{XNOR}_{\text{AIL}}$ activation function to significantly outperform an identical model utilizing the ReLU activation function.

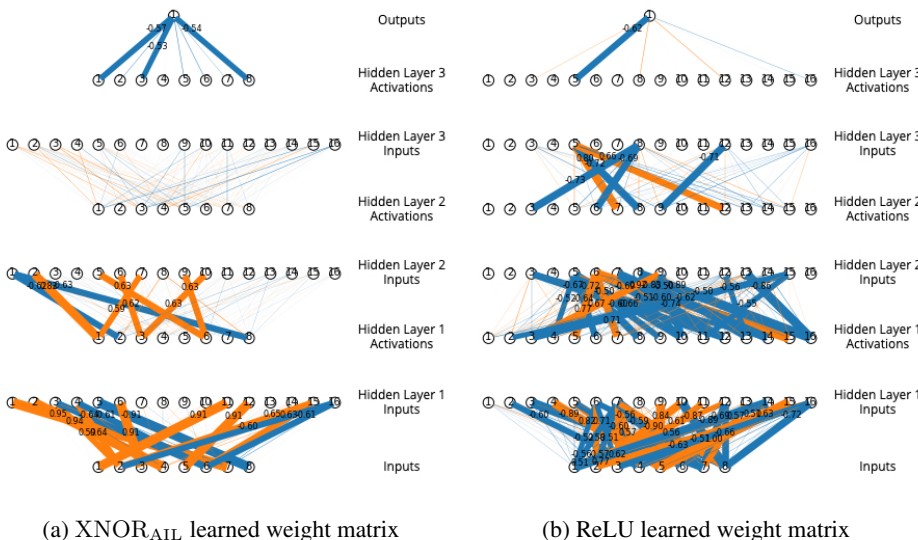

(a) $\text{XNOR}_{\text{AIL}}$ learned weight matrix      (b) ReLU learned weight matrix

Figure 15: Training results, regression experiment on second synthetic dataset

We found that a simple model with three hidden layers, each with eight neurons, utilizing $\text{XNOR}_{\text{AIL}}$ was able to go from a validation RMSE of 0.287 at the beginning of training to a validation RMSE of 0.016 after 100 epochs. Comparatively, an identical model utilizing the ReLU activation function was only able to achieve a validation RMSE of 0.271 after 100 epochs. In order for our ReLU network to match the validation RMSE of our 8-neuron-per-layer $\text{XNOR}_{\text{AIL}}$ model, we had to increase the model size by 32 times to 256 neurons at each hidden layer.

---

[2]We hypothesize that, because our $\text{XNOR}_{\text{AIL}}$ activation function reduces the number of hidden layer neurons by a factor of $k$, having a reduced number of neurons at each layer creates a bottleneck in the later layers of the network which restricts the amount of information that made its way through to the final layer

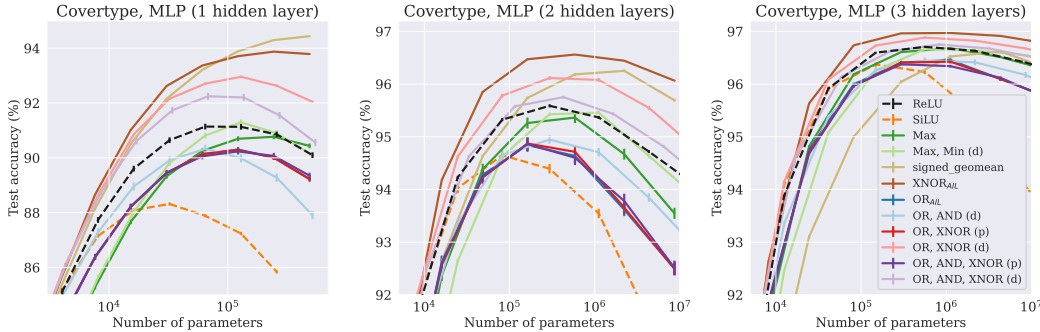

Figure 16: We trained MLPs on the Covertype dataset, with a fixed 80:20 random split. Trained with Adam, 50 ep., 1-cycle, using LRs determined automatically with LR-finder. Mean (bars: std dev) of $n=10$ weight inits.

## A.8 COVERTYPE

Our one, two, and three-layer MLPs were trained using 1-cycle (Smith & Topin, 2017; Smith, 2018) for 50 epochs, with a batch size of $1024$, without weight decay. We used a subset of 15% of the training data for validation. The learning rate was selected using an automated learning rate finder approach, verified against the validation partition. For our final experiments, the validation partition was included in the training data. No data augmentation was performed.

## A.9 BACH CHORALE TRAINING DETAILS

For each of the 4 voices, we restricted the available pitches to 3 octaves, resulting in 37 one-hot tokens (including silence). Since we only planned on feeding small time-windows of the chorale into our model, for pre-processing, we converted the data into a shape of $(\texttt{seq\_len}, 4, 37)$, where we set `seq_len` to 4.

To generate training examples for the discriminator, we transposed by $\{-5, -4, \ldots, 5, 6\}$ semitones, chosen uniformly at random, and there was a $0.5$ probability that the sample was corrupted by the following method:

- Choose 2-3 notes in the $(\texttt{seq\_len} \times 4)$ window to be corrupted
- For each note, corrupt by the following mixture distribution:
    - $(p = 0.6)$ Sample a pitch from a Gaussian centered on the existing note, with $\sigma = 3$ semitones, forcing the new pitch to be distinct
    - $(p = 0.2)$ Copy a pitch from the current voice, forcing the new pitch to be distinct
    - $(p = 0.2)$ Extend the previous note in time
    - $(p = 0.1)$ Sample uniformly from all 37 possible tokens

## A.10 CORRELATIONS BETWEEN PRE-ACTIVATIONS

Results on correlations between weights in the JSB Chorale models are shown in Figure 18. We found that when taking all pre-activations into account, every activation function generally showed independence between features. Interestingly, the cosine similarities between inputs that were paired together for the bivariate activation functions showed anticorrelation in almost all cases where Max or $\text{OR}_{\text{AIL}}$ were used, and other cases generally showed more correlation than ReLU.

We found that randomly selected pairs of preactivation features within the same layer have correlations that are given by a Gaussian-like distribution centered around zero. This was the case for all of the activation functions we tested. The behaviour of randomly selected pairs of features is thus reasonably consistent with the assumption of independence which we have made. We also investigated the correlation between the pairs of preactivation features which were passing into our two-dimensional activation functions. Here, we found the correlation structure is different, and the correlation depends

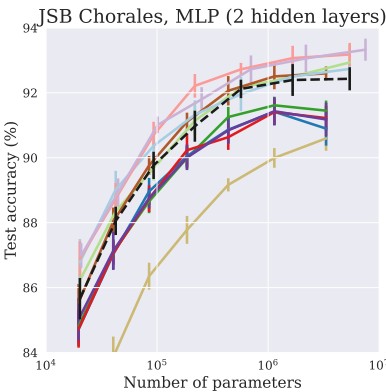

Figure 17: We trained 2-layer MLPs discriminators on JSB Chorales using Adam (constant LR $1 \times 10^{-3}$, 150 ep.), Mean (bars: std dev) of $n = 10$ weight inits.

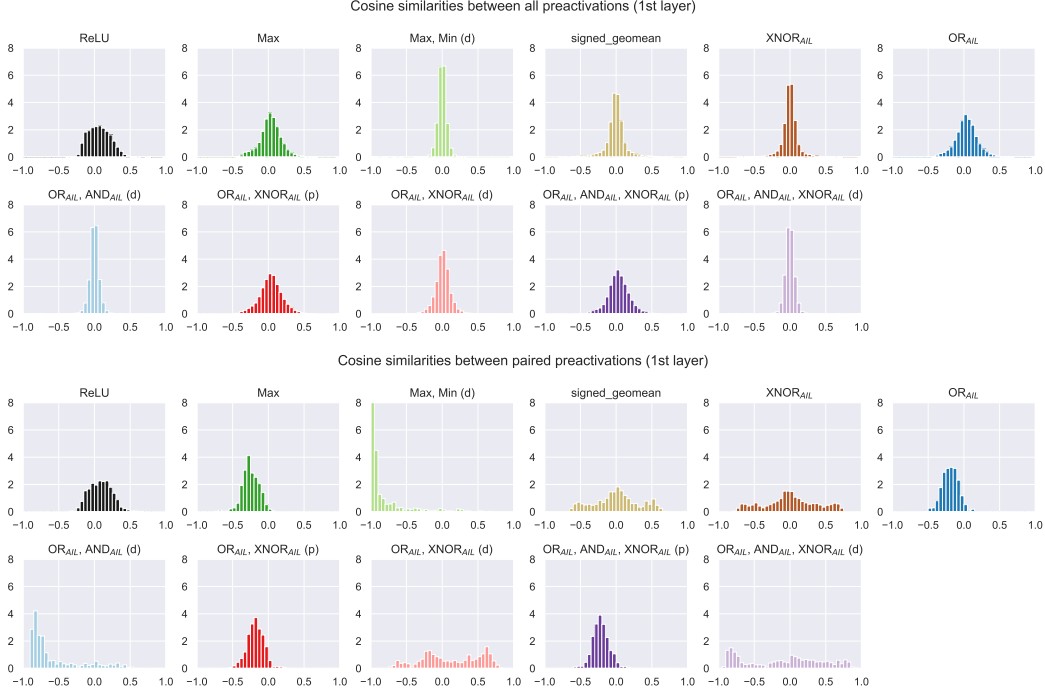

Figure 18: Cosine similarities between pre-activation weights of two activation functions in the first layer of an MLP trained on JSB Chorales.

on the activation function being used. With Max and $OR_{AIL}$ activations, the network learns to make the columns of the weight matrix (and hence the preactivation scores for the pair of features) be inversely correlated. With $XNOR_{AIL}$, the network learns features which are either positively or negatively correlated (a wider distribution of correlations than seen with random pairs of features). We observe that (in all cases) the network learns to make the features passed to the AIL activation functions be correlated instead of independent, despite our assumption of independence. So it appears clear that the assumption of independence is violated, but also that it doesn't *really* matter because the network is choosing to break the assumption and induce these correlations between the features to get better performance.

## A.11 CNN AND MLP ON MNIST

In this experiment we trained MLP and CNN models for 10 epochs on the MNIST dataset using Adam optimizer, one-cycle learning rate schedule, and cross entropy loss, with batch size of 256. We augmented our training samples using a random affine transformation from: rotation of $\pm 10$ degrees, scale of factor $0.8$ to $1.2$, translation with max absolute fraction for horizontal and vertical directions of $0.08$, and shear parallel to the x-axes of $\pm 0.3$.

The hyper-parameters for the optimizer and scheduler were selected through a random search of the hyper-parameter space. We chose to do random search instead of grid search because it typically yields better results for the same number of test cases.

For the hyper-parameter search, we trained on the first $50\,000$ samples of the training partition and used the final $10\,000$ samples as a validation set. We ran the search for four iterations, with each iteration sampling 120 different hyper-parameter settings. The initial bounds for our hyper-parameter samples were set as described in Table 3. These bounds were chosen to be suitably wide such that the optimal configuration should be contained within them for all activation functions considered.

Table 3: Hyper-parameter random search parameters.

| Hyper-parameter | Variable | Sampling (initial bounds) |
|---|---|---|
| Adam beta1 | $1 - 10^x$ | $x \sim \text{Uniform}(-3, -0.5)$ |
| Adam beta2 | $1 - 10^x$ | $x \sim \text{Uniform}(-5, -1)$ |
| Adam epsilon | $10^x$ | $x \sim \text{Uniform}(-10, -6)$ |
| Weight decay | $10^x$ | $x \sim \text{Uniform}(-7, -3)$ |
| One-cycle max LR | $10^x$ | $x \sim \text{Uniform}(-4, 0)$ |
| One-cycle peak | $x$ | $x \sim \text{Uniform}(0.1, 0.5)$ |

The bounds on our uniform random variable $x$ were tightened at each iteration by selecting the top-5 performing hyper-parameter settings, taking the mean $\bar{x}$ and weighted standard deviation $\sigma$ across those settings, and re-setting the bounds for the next iteration to be equal to $\bar{x} \pm 1.5\,\sigma$. After the fourth iteration, we ran a final iteration where we selected the top-10 performing hyper-parameter settings across the four previous iterations, and then re-ran these for another 120 seeds (i.e. random weight initializations). We then selected the hyper-parameter settings which had the highest performance across all 120 seeds.

We found that the XNOR activation function required quite different hyper-parameters than the other activation functions, but OR and Max activation functions used similar hyper-parameters to ReLU. However, we have no reason to believe that the proposed AIL activation functions are more susceptible than others to the choice of hyper-parameters or the way in which hyper-parameters are selected.

Our MLP model consisted of two hidden layers of equal size with batch norm applied to each. The number of neurons in each layer was set, taking into consideration the current activation function being tested, to ensure the number of trainable parameters in the network remained static across experiments

Our CNN model consisted of six layers, where each layer was comprised of a set of 2D convolution filters (kernel size 3, stride 1, padding 1), batch normalization, and a non-linear activation. The network also applied a pooling layer (kernel size 2, stride 2) at the end of every second layer. After the 6 CNN layers the network flattens the output and applies three linear layers. The number of output channels (ie. pre-activations) for the six 2D convolutions were $[c, 2c, 4c, 4c, 8c, 8c]$, and the number of pre-activation neurons produced by the subsequent linear layers were $[32c, 16c]$. Similar to the MLP model, $c$ was chosen to ensure the number of trainable parameters in the network remained fixed across experiments. When varying the number of network parameters in our experiments, if the number of parameters dropped below $1 \times 10^6$ in the CNN model, we changed the structure of the convolution layers and linear layers to $[c, c, c, c, 1.5c, 1.5c]$ and $[2c, 1.5c]$, respectively. This ensured that each layer had at least two channels for our activation functions to aggregate.

Once our optimal hyper-parameters were selected for both our MLP and CNN models, we then trained each model several times with varying number of trainable parameters. The reason we vary

the number of parameters in the network instead of the network size/structure is because a network using our logical activation functions can have a significantly different number of parameters than an identical network using ReLU activation, because our logical activations aggregate across neurons at each layer.

## A.12 RESNET50 ON CIFAR-10/100

For this experiment we trained a ResNet50 model for 100 epochs on CIFAR-10 and CIFAR-100 using Adam optimizer, one-cycle learning rate schedule, cross entropy loss, and augmentations derived for CIFAR-10 by AutoAugment (Cubuk et al., 2018), with batch size of 128. The hyper-parameters for the optimizer and scheduler were determined through a random search of the hyper parameter space on the CIFAR-100 dataset[3]. The hyper-parameter search was tuned against a partition of 10% of the CIFAR-100 training samples. Our hyper-parameter search follows a similar approach to the one used for our MLP and CNN models on MNIST (Appendix A.11), but due to compute constraints with our larger models here we only trained 100 seeds each iteration, and only re-examined the top 10 seeds in the final round.

For PReLU, we did not perform a hyper-parameter search and merely re-used the same hyper-parameters as discovered in the search with ReLU.

Once our optimal hyper-parameters were selected we trained a ResNet50 model with automatic mixed precision using four different width parameters: 0.5, 1, 2, and 4. Because activation functions such as $XNOR_{AIL}$ reduce the number of post-activation neurons by a factor of 2, and the $\{OR, AND, XNOR_{AIL} (d)\}$ activation function increases the number of post-activation neurons by a factor of $3/2$, we adjust the width parameter for these activation function to result in all networks having approximately the same number of parameters. For the $2 \rightarrow 1$ activation functions (Max, $OR_{AIL}$, $XNOR_{AIL}$, $\{OR, XNOR_{AIL} (p)\}$, and $\{OR, AND, XNOR_{AIL} (p)\}$) we use width values of 0.75, 1.5, 3, and 6. For the $2 \rightarrow 3$ activation function $\{OR, AND, XNOR_{AIL} (d)\}$, we use width values of 0.4, 0.75, 1.5, and 3.

## A.13 TRANSFER LEARNING

We used a pretrained ResNet50 model taken from the pytorch hub. The 2-layer MLP head was trained for 25 epochs on each dataset. Since we are interested in transfer learning in a data-limited regime for these experiments, we used only a small amount of data augmentation. We applied horizontal flip ($p = 0.5$), scaling (0.7 to 1), aspect ratio stretching ($3/4$ to $4/3$), and colour jitter (intensity 0.4) only. Pixel intensity normalization was done against the ImageNet mean and std.

The MLP head was optimized using SGD, momentum 0.9. We used a batch size of 128, maximum learning rate 0.01, and weight decay $1 \times 10^{-4}$. Before training began, we passed one epoch worth of inputs through the network without updating the weights in order to refresh the batch normalization statistics to the new dataset. We also performed one epoch of training with a warmup learning rate of $1 \times 10^{-5}$ before commencing training at the maximum learning rate of 0.01. The learning rate was decayed with a cosine annealing schedule over 24 epochs. We report the performance of the final model at the end of the 25 epochs.

We used a pre-activation width of 512 neurons for ReLU and other and activation functions which map $1 \rightarrow 1$. To approximately match up the number of parameters, we used a width of 650 for activation functions which map $2 \rightarrow 1$, and 438 for $\{OR, AND, XNOR_{AIL} (d)\}$ which maps features $3 \rightarrow 2$. These values control the total number of parameters in the head to be the same for datasets with 100 classes (the median number of classes in the datasets we considered). Our results with widths 438/512/650 and $\sim$600k parameters are shown in Table 1. The precise widths and number of trainable parameters used for these experiments are shown in Table 4.

Since the transfer learning experiments were performed without retraining the base network, we found (Table 1) the performance is limited by the features which the base network produces, whose relevance depends on the other lap between the domain of the pretraining task (ImageNet) and the new task. This information bottleneck, and variation in its utility, means the variation between

---

[3]We used the same set of hyper-parameters from this search for both the CIFAR-10 and CIFAR-100 experiments.

performance on different datasets is much larger than the variation for different activation functions within the same dataset.

For transfer tasks which involve coarse-grained discrimination on images which are similar to ImageNet, the embedding generated by the pretrained model is already sufficient to separate the classes in the new dataset. Examples of this are Caltech101, where linear layer beats out using additional layers with non-linearities; and STL-10, where our best model is on-par with the linear model. The results on these two transfer learning tasks are too simple to be of interest to study. The performance is limited by the information retained by the pretrained embedding, but it appears that what task-relevant information is there is readily available with a linear layer without needing additional logic to interpret it.

Other transfer learning tasks we attempted are less trivial and show a larger difference in performance across the activation functions, and a gap from the linear layer readout model. The Stanford Cars dataset involves fine-grained discrimination between different car models, for which features generated by a model pretrained on ImageNet are not effective. The SVHN dataset, which contains images of house numbers, is coarse-grained but uses images which are outside the domain of ImageNet, which makes the transfer-learning task more difficult. Additionally, our use of random horizontal reflections during training will have hindered performance on this dataset by notably increasing the task complexity, since numerals are not invariant under reflection.

Table 4: Hidden pre-activation widths and number of trainable parameters used in transfer learning experiments (corresponding to results shown in Table 1).

| Activation function | Map | Width | № Trainable Parameters | | | | | | |
| --- | --- | --- | --- | --- | --- | --- | --- | --- | --- |
| | | | Caltech101 | CIFAR10 | CIFAR100 | Flowers | Cars | STL-10 | SVHN |
| Linear layer only | | | 52k | 5k | 51k | 52k | 101k | 5k | 5k |
| ReLU | $1\rightarrow1$ | 512 | 577k | 530k | 577k | 578k | 626k | 530k | 530k |
| LeakyReLU | $1\rightarrow1$ | 512 | 577k | 530k | 577k | 578k | 626k | 530k | 530k |
| PReLU | $1\rightarrow1$ | 512 | 577k | 530k | 577k | 578k | 626k | 530k | 530k |
| Softplus | $1\rightarrow1$ | 512 | 577k | 530k | 577k | 578k | 626k | 530k | 530k |
| ELU | $1\rightarrow1$ | 512 | 577k | 530k | 577k | 578k | 626k | 530k | 530k |
| CELU | $1\rightarrow1$ | 512 | 577k | 530k | 577k | 578k | 626k | 530k | 530k |
| SELU | $1\rightarrow1$ | 512 | 577k | 530k | 577k | 578k | 626k | 530k | 530k |
| GELU | $1\rightarrow1$ | 512 | 577k | 530k | 577k | 578k | 626k | 530k | 530k |
| SiLU | $1\rightarrow1$ | 512 | 577k | 530k | 577k | 578k | 626k | 530k | 530k |
| Hardswish | $1\rightarrow1$ | 512 | 577k | 530k | 577k | 578k | 626k | 530k | 530k |
| Mish | $1\rightarrow1$ | 512 | 577k | 530k | 577k | 578k | 626k | 530k | 530k |
| Softsign | $1\rightarrow1$ | 512 | 577k | 530k | 577k | 578k | 626k | 530k | 530k |
| Tanh | $1\rightarrow1$ | 512 | 577k | 530k | 577k | 578k | 626k | 530k | 530k |
| GLU | $2\rightarrow1$ | 650 | 578k | 549k | 578k | 579k | 609k | 549k | 549k |
| Max | $2\rightarrow1$ | 650 | 578k | 549k | 578k | 579k | 609k | 549k | 549k |
| Max, Min (d) | $1\rightarrow1$ | 512 | 577k | 530k | 577k | 578k | 626k | 530k | 530k |
| $\text{XNOR}_{\text{AIL}}$ | $2\rightarrow1$ | 650 | 578k | 549k | 578k | 579k | 609k | 549k | 549k |
| $\text{OR}_{\text{AIL}}$ | $2\rightarrow1$ | 650 | 578k | 549k | 578k | 579k | 609k | 549k | 549k |
| OR, $\text{AND}_{\text{AIL}}$ (d) | $1\rightarrow1$ | 512 | 577k | 530k | 577k | 578k | 626k | 530k | 530k |
| OR, $\text{XNOR}_{\text{AIL}}$ (p) | $2\rightarrow1$ | 650 | 578k | 549k | 578k | 579k | 609k | 549k | 549k |
| OR, $\text{XNOR}_{\text{AIL}}$ (d) | $1\rightarrow1$ | 512 | 577k | 530k | 577k | 578k | 626k | 530k | 530k |
| OR, AND, $\text{XNOR}_{\text{AIL}}$ (p) | $2\rightarrow1$ | 650 | 578k | 549k | 578k | 579k | 609k | 549k | 549k |
| OR, AND, $\text{XNOR}_{\text{AIL}}$ (d) | $2\rightarrow3$ | 438 | 579k | 519k | 579k | 580k | 642k | 519k | 519k |

Additionally, we ran the experiment again with a width of $w = 512$ for the $2\rightarrow1$ activation functions. The results with constant pre-activation width $w = 512$ are shown in Table 5. By using a constant pre-activation width, the number of parameters used is not consistent across activation functions. The number of trainable parameters in each experiment is shown in tab:transfer-nparam-512.

We found that reducing the width from 650 to 512 (and hence reducing total number of trainable parameters) reduced the performance of $2\rightarrow1$ activation functions $\text{XNOR}_{\text{AIL}}$, $\text{OR}_{\text{AIL}}$, {OR, $\text{XNOR}_{\text{AIL}}$ (p)}, and {OR, AND, $\text{XNOR}_{\text{AIL}}$ (p)}. Increasing the width from 438 to 512 increased the performance of {OR, AND, $\text{XNOR}_{\text{AIL}}$ (d)}.

Table 5: Transfer learning with from a frozen ResNet-18 architecture pretrained on ImageNet-1k to other computer vision datasets. As with Table 1, but in this case we show results where the MLP has a pre-activation width of $w = 512$. Note that although the pre-activation width is constant, the number of parameters in the network is not consistent between experiments. The number of parameters is shown in Table 6. Mean (standard error) of $n = 5$ inits of the MLP (same pretrained network).

| Activation function | Test Accuracy (%) | | | | | | |
|---|---|---|---|---|---|---|---|
| | Caltech101 | CIFAR10 | CIFAR100 | Flowers | Cars | STL-10 | SVHN |
| Linear layer only | $88.35_{\pm0.15}$ | $78.56_{\pm0.09}$ | $57.39_{\pm0.09}$ | $92.32_{\pm0.20}$ | $33.51_{\pm0.06}$ | $94.68_{\pm0.02}$ | $45.42_{\pm0.06}$ |
| ReLU | $86.58_{\pm0.17}$ | $81.63_{\pm0.05}$ | $58.04_{\pm0.11}$ | $90.71_{\pm0.26}$ | $30.97_{\pm0.26}$ | $94.62_{\pm0.06}$ | $51.39_{\pm0.06}$ |
| LeakyReLU | $86.60_{\pm0.13}$ | $81.67_{\pm0.11}$ | $58.01_{\pm0.09}$ | $90.73_{\pm0.32}$ | $31.09_{\pm0.24}$ | $94.61_{\pm0.05}$ | $51.40_{\pm0.05}$ |
| PReLU | $87.83_{\pm0.21}$ | $81.03_{\pm0.13}$ | $58.90_{\pm0.18}$ | $93.17_{\pm0.19}$ | $39.84_{\pm0.18}$ | $94.54_{\pm0.05}$ | $51.42_{\pm0.09}$ |
| Softplus | $86.16_{\pm0.18}$ | $79.13_{\pm0.08}$ | $56.58_{\pm0.07}$ | $89.39_{\pm0.29}$ | $21.23_{\pm0.13}$ | $94.63_{\pm0.03}$ | $47.44_{\pm0.06}$ |
| ELU | $87.18_{\pm0.09}$ | $80.44_{\pm0.08}$ | $58.08_{\pm0.10}$ | $91.71_{\pm0.14}$ | $34.70_{\pm0.06}$ | $94.55_{\pm0.05}$ | $50.07_{\pm0.07}$ |
| CELU | $87.18_{\pm0.09}$ | $80.44_{\pm0.08}$ | $58.08_{\pm0.10}$ | $91.71_{\pm0.14}$ | $34.70_{\pm0.06}$ | $94.55_{\pm0.05}$ | $50.07_{\pm0.07}$ |
| SELU | $87.74_{\pm0.09}$ | $79.93_{\pm0.13}$ | $58.24_{\pm0.06}$ | $92.27_{\pm0.13}$ | $37.51_{\pm0.17}$ | $94.53_{\pm0.07}$ | $49.38_{\pm0.06}$ |
| GELU | $87.10_{\pm0.15}$ | $81.39_{\pm0.09}$ | $58.51_{\pm0.13}$ | $91.51_{\pm0.15}$ | $33.43_{\pm0.15}$ | $94.62_{\pm0.06}$ | $51.56_{\pm0.08}$ |
| SiLU | $86.91_{\pm0.11}$ | $80.53_{\pm0.11}$ | $58.14_{\pm0.12}$ | $91.37_{\pm0.18}$ | $32.15_{\pm0.17}$ | $94.59_{\pm0.05}$ | $50.69_{\pm0.09}$ |
| Hardswish | $87.12_{\pm0.12}$ | $80.10_{\pm0.10}$ | $58.25_{\pm0.10}$ | $91.56_{\pm0.25}$ | $33.17_{\pm0.23}$ | $94.62_{\pm0.05}$ | $50.09_{\pm0.09}$ |
| Mish | $87.11_{\pm0.12}$ | $81.09_{\pm0.11}$ | $58.37_{\pm0.10}$ | $91.61_{\pm0.15}$ | $33.75_{\pm0.14}$ | $94.61_{\pm0.05}$ | $51.29_{\pm0.08}$ |
| Softsign | $81.47_{\pm0.18}$ | $80.03_{\pm0.09}$ | $54.84_{\pm0.09}$ | $82.34_{\pm0.22}$ | $17.33_{\pm0.10}$ | $94.70_{\pm0.03}$ | $49.48_{\pm0.07}$ |
| Tanh | $87.48_{\pm0.06}$ | $80.56_{\pm0.07}$ | $57.35_{\pm0.08}$ | $90.32_{\pm0.20}$ | $29.51_{\pm0.12}$ | $94.63_{\pm0.07}$ | $50.15_{\pm0.08}$ |
| GLU | $86.34_{\pm0.16}$ | $79.35_{\pm0.05}$ | $57.22_{\pm0.12}$ | $90.10_{\pm0.20}$ | $26.72_{\pm0.13}$ | $94.63_{\pm0.05}$ | $48.51_{\pm0.09}$ |
| Max | $86.86_{\pm0.11}$ | $81.56_{\pm0.06}$ | $58.12_{\pm0.10}$ | $90.59_{\pm0.25}$ | $32.80_{\pm0.04}$ | $94.65_{\pm0.05}$ | $51.15_{\pm0.08}$ |
| Max, Min (d) | $87.23_{\pm0.13}$ | $82.31_{\pm0.10}$ | $59.05_{\pm0.10}$ | $91.68_{\pm0.18}$ | $34.91_{\pm0.12}$ | $94.64_{\pm0.04}$ | $51.72_{\pm0.04}$ |
| XNOR$_{AIL}$ | $86.42_{\pm0.13}$ | $81.74_{\pm0.05}$ | $57.88_{\pm0.09}$ | $90.50_{\pm0.17}$ | $31.55_{\pm0.11}$ | $94.78_{\pm0.04}$ | $51.23_{\pm0.10}$ |
| OR$_{AIL}$ | $87.28_{\pm0.09}$ | $81.81_{\pm0.02}$ | $58.68_{\pm0.05}$ | $91.78_{\pm0.23}$ | $35.28_{\pm0.09}$ | $94.67_{\pm0.07}$ | $51.27_{\pm0.12}$ |
| OR, AND$_{AIL}$ (d) | $87.43_{\pm0.11}$ | $82.38_{\pm0.06}$ | $59.90_{\pm0.08}$ | $92.07_{\pm0.18}$ | $37.16_{\pm0.15}$ | $94.55_{\pm0.05}$ | $52.11_{\pm0.09}$ |
| OR, XNOR$_{AIL}$ (p) | $87.26_{\pm0.06}$ | $81.85_{\pm0.06}$ | $58.72_{\pm0.09}$ | $91.80_{\pm0.24}$ | $35.27_{\pm0.09}$ | $94.64_{\pm0.07}$ | $51.25_{\pm0.14}$ |
| OR, XNOR$_{AIL}$ (d) | $87.09_{\pm0.21}$ | $82.20_{\pm0.04}$ | $59.44_{\pm0.07}$ | $91.90_{\pm0.10}$ | $36.88_{\pm0.10}$ | $94.69_{\pm0.06}$ | $52.02_{\pm0.16}$ |
| OR, AND, XNOR$_{AIL}$ (p) | $87.03_{\pm0.19}$ | $81.75_{\pm0.07}$ | $58.72_{\pm0.08}$ | $91.85_{\pm0.14}$ | $35.46_{\pm0.14}$ | $94.69_{\pm0.04}$ | $51.32_{\pm0.09}$ |
| OR, AND, XNOR$_{AIL}$ (d) | $87.26_{\pm0.15}$ | $82.45_{\pm0.08}$ | $60.20_{\pm0.11}$ | $92.41_{\pm0.17}$ | $37.89_{\pm0.14}$ | $94.65_{\pm0.04}$ | $52.19_{\pm0.09}$ |

Table 6: Number of trainable parameters used in transfer learning experiments (corresponding to results shown in Table 5).

| Activation function | Map | Width | № Trainable Parameters | | | | | | |
|---|---|---|---|---|---|---|---|---|---|
| | | | Caltech101 | CIFAR10 | CIFAR100 | Flowers | Cars | STL-10 | SVHN |
| Linear layer only | | | 52k | 5k | 51k | 52k | 101k | 5k | 5k |
| ReLU | $1 \to 1$ | 512 | 577k | 530k | 577k | 578k | 626k | 530k | 530k |
| LeakyReLU | $1 \to 1$ | 512 | 577k | 530k | 577k | 578k | 626k | 530k | 530k |
| PReLU | $1 \to 1$ | 512 | 577k | 530k | 577k | 578k | 626k | 530k | 530k |
| Softplus | $1 \to 1$ | 512 | 577k | 530k | 577k | 578k | 626k | 530k | 530k |
| ELU | $1 \to 1$ | 512 | 577k | 530k | 577k | 578k | 626k | 530k | 530k |
| CELU | $1 \to 1$ | 512 | 577k | 530k | 577k | 578k | 626k | 530k | 530k |
| SELU | $1 \to 1$ | 512 | 577k | 530k | 577k | 578k | 626k | 530k | 530k |
| GELU | $1 \to 1$ | 512 | 577k | 530k | 577k | 578k | 626k | 530k | 530k |
| SiLU | $1 \to 1$ | 512 | 577k | 530k | 577k | 578k | 626k | 530k | 530k |
| Hardswish | $1 \to 1$ | 512 | 577k | 530k | 577k | 578k | 626k | 530k | 530k |
| Mish | $1 \to 1$ | 512 | 577k | 530k | 577k | 578k | 626k | 530k | 530k |
| Softsign | $1 \to 1$ | 512 | 577k | 530k | 577k | 578k | 626k | 530k | 530k |
| Tanh | $1 \to 1$ | 512 | 577k | 530k | 577k | 578k | 626k | 530k | 530k |
| GLU | $2 \to 1$ | 512 | 420k | 397k | 420k | 420k | 445k | 397k | 397k |
| Max | $2 \to 1$ | 512 | 420k | 397k | 420k | 420k | 445k | 397k | 397k |
| Max, Min (d) | $1 \to 1$ | 512 | 577k | 530k | 577k | 578k | 626k | 530k | 530k |
| XNOR$_{AIL}$ | $2 \to 1$ | 512 | 420k | 397k | 420k | 420k | 445k | 397k | 397k |
| OR$_{AIL}$ | $2 \to 1$ | 512 | 420k | 397k | 420k | 420k | 445k | 397k | 397k |
| OR, AND$_{AIL}$ (d) | $1 \to 1$ | 512 | 577k | 530k | 577k | 578k | 626k | 530k | 530k |
| OR, XNOR$_{AIL}$ (p) | $2 \to 1$ | 512 | 420k | 397k | 420k | 420k | 445k | 397k | 397k |
| OR, XNOR$_{AIL}$ (d) | $1 \to 1$ | 512 | 577k | 530k | 577k | 578k | 626k | 530k | 530k |
| OR, AND, XNOR$_{AIL}$ (p) | $2 \to 1$ | 512 | 420k | 397k | 420k | 420k | 445k | 397k | 397k |
| OR, AND, XNOR$_{AIL}$ (d) | $2 \to 3$ | 512 | 734k | 664k | 733k | 735k | 807k | 664k | 664k |

### A.14 ABSTRACT REASONING

Abstract reasoning is challenging for neural networks to learn because their structure and objective function are more effective for tasks which come instinctively to humans ("System 1" of Kahneman, 2011), such as object recognition, as opposed to demanding ("System 2") logic tasks.

In recent years, several challenges have been proposed to evaluate the ability of neural networks to perform abstract reasoning. The Raven's Progressive Matrices (Raven & Court, 1938) are a long-standing IQ test, which have been emulated by Barrett et al. (2018) with Procedurally Generated Matrices (PGM), and then the RAVEN task by (Zhang et al., 2019). Recently Hu et al. (2020) and Benny et al. (2021) have improved on that task with I-RAVEN and RAVEN-FAIR, respectively, both aiming to make the task more balanced. Other abstract reasoning tasks have also been proposed (Fleuret et al., 2011; Johnson et al., 2017; Barrett et al., 2018).

We considered the application of AIL activation functions in the context of the I-RAVEN task, by adapting the Stratified Rule-Aware Network (SRAN) of Hu et al. (2020) to include our AIL activation functions. We first added LayerNorm to the network, and then swapped out the seven ReLU activations in the gating module. The architecture for the three ResNet-18 (He et al., 2015) base models were unchanged. Where necessary, the number of units per layer was modified to facilitate the change in dimensionality caused by our activation function. The networks were trained using the same procedure as described by Hu et al. (2020).

Table 7: Performance of SRAN-based models on the I-RAVEN dataset (Hu et al., 2020). Bold: best. Underlined: second best. Background: color scale from worst in to best, linear with accuracy value.

| Activation function | Params | I-RAVEN Test Acc (%) | | | | | | | |
|---|---|---|---|---|---|---|---|---|---|
| | | Acc | Center | 2×2G | 3×3G | O-IC | O-IG | L-R | U-D |
| ReLU, Base SRAN (Hu et al., 2020) | 44.0M | 60.8 | 78.2 | **50.1** | 42.4 | 68.2 | 46.3 | 70.1 | 70.3 |
| ReLU, SRAN+LayerNorm | 45.6M | 63.0 | 84.0 | **50.0** | 42.5 | 69.9 | 48.3 | 73.5 | 72.3 |
| PReLU | 45.6M | 54.5 | 69.2 | 45.4 | 40.5 | 64.4 | 47.6 | 57.5 | 57.3 |
| CELU | 45.6M | 56.6 | 73.5 | 46.5 | 41.0 | 66.6 | 46.1 | 62.8 | 59.7 |
| SELU | 45.6M | 53.5 | 67.2 | 43.8 | 40.1 | 62.8 | 44.5 | 58.0 | 58.1 |
| GELU | 45.6M | 61.4 | 80.8 | 49.2 | 42.5 | 69.2 | 48.3 | 70.3 | 69.2 |
| SiLU | 45.6M | 59.4 | 78.0 | 47.2 | 41.4 | 66.3 | 47.8 | 69.1 | 66.2 |
| Max | 44.6M | 57.8 | 76.3 | 45.2 | 39.7 | 65.3 | 48.7 | 64.6 | 64.7 |
| Max, Min (d) | 45.6M | 60.2 | 80.2 | 49.5 | 41.9 | 66.5 | 47.0 | 68.5 | 67.7 |
| XNOR$_{AIL}$ | 44.6M | 57.7 | 74.7 | 46.0 | 40.0 | 66.8 | 47.7 | 65.6 | 63.2 |
| OR$_{AIL}$ | 44.6M | **64.3** | **84.4** | 49.5 | **44.0** | **71.5** | 47.1 | **76.5** | **77.0** |
| OR, AND$_{AIL}$ (d) | 45.6M | 57.5 | 74.7 | 45.6 | 41.3 | 68.3 | 44.9 | 64.5 | 63.0 |
| OR, XNOR$_{AIL}$ (p) | 44.6M | 59.8 | 80.5 | 45.8 | 41.4 | 67.2 | 48.2 | 67.5 | 68.1 |
| OR, XNOR$_{AIL}$ (d) | 45.6M | 62.8 | 83.7 | 49.1 | 43.3 | 68.1 | 49.5 | 73.8 | 72.2 |
| OR, AND, XNOR$_{AIL}$ (p) | 44.6M | 55.0 | 68.2 | 47.2 | 41.1 | 65.0 | 45.0 | 61.0 | 57.5 |

We found the network using OR$_{AIL}$ activation function performed best overall, and across most of the subtasks. The second best performing activation functions were {OR$_{AIL}$, XNOR$_{AIL}$ (d)} and ReLU.

### A.15 COMPOSITIONAL ZERO-SHOT LEARNING

Zero-shot learning encompasses all problems which involve completing novel tasks which the subject has never seen before. The subject must infer both the task and its solution based on their previous experiences (a meta-learning task). Compositional zero-shot learning is a subset of zero-shot learning which involves combining knowledge about multiple stimulus properties together in novel pairings. For instance, if the network has been trained on "sliced bread," "sliced pear," and "caramelized pear," is it able to classify images of "caramelized bread" despite having never seen an example of this before?

We based our experiments on the Task-driven Modular Networks (TMN) proposed by Purushwalkam et al. (2019). We used the code shared by the authors, but were unable to replicate the results they reported in the paper when using a different random seed (see Table 8). We adapted this network by changing out all the ReLU activation functions in the gate and module networks with a different

activation function. Because the modules each terminate with an activation function, we needed to double the size of the hidden layer for some of our networks in order to maintain the dimensionality of the output. Consequently, some experiments had around 50% more parameters in total than others.

Experiments were performed on the MIT-States dataset (Isola et al., 2015).We trained and tested on the corresponding partitions of the dataset as introduced by Purushwalkam et al. (2019). We used the same paradigm as Purushwalkam et al. (2019): Adam (Kingma & Ba, 2017) with a learning rate of 0.001 for the module network and 0.01 for the gating network, momentum 0.9, batch size 256, weight decay $5 \times 10^{-5}$. We evaluated the network using the AUC between seen and unseen samples (Purushwalkam et al., 2019). The network was trained until the validation AUC had plateaued, determined by not increasing for 5 epochs. We selected the model from the epoch with highest validation AUC to apply to the test set. The best performance was typically attained after around 5 epochs.

As shown in Table 8, we find that the trio of activation functions applied in parallel, $\{\mathrm{OR}, \mathrm{AND}, \mathrm{XNOR}_{\mathrm{AIL}} \text{ (p)}\}$, performs best.

Table 8: Performance of TMN-based networks at compositional zero-shot learning (CZSL) on the MIT-States dataset. Mean (standard error) of $n = 5$ random initializations.

| Activation function | Mapping | Params | MIT-States Test AUC (%) | | |
| --- | --- | --- | --- | --- | --- |
| | | | Top-1 | Top-2 | Top-3 |
| TMN (Purushwalkam et al., 2019) | $1 \rightarrow 1$ | | 2.9 | 7.1 | 11.5 |
| TMN repro. (ReLU) | $1 \rightarrow 1$ | 438 k | $2.47 \pm 0.07$ | $6.42 \pm 0.09$ | $10.27 \pm 0.11$ |
| Max | $2 \rightarrow 1$ | 650 k | $2.42 \pm 0.07$ | $6.37 \pm 0.08$ | $10.28 \pm 0.06$ |
| Max, Min (d) | $1 \rightarrow 1$ | 438 k | $2.53 \pm 0.06$ | $6.69 \pm 0.11$ | $10.61 \pm 0.14$ |
| $\mathrm{XNOR}_{\mathrm{AIL}}$ | $2 \rightarrow 1$ | 650 k | $1.22 \pm 0.05$ | $3.47 \pm 0.12$ | $5.82 \pm 0.19$ |
| $\mathrm{OR}_{\mathrm{AIL}}$ | $2 \rightarrow 1$ | 650 k | $2.65 \pm 0.05$ | $6.80 \pm 0.08$ | $10.78 \pm 0.13$ |
| $\mathrm{OR}, \mathrm{AND}_{\mathrm{AIL}}$ (d) | $1 \rightarrow 1$ | 438 k | $2.61 \pm 0.05$ | $6.73 \pm 0.05$ | $10.77 \pm 0.12$ |
| $\mathrm{OR}, \mathrm{XNOR}_{\mathrm{AIL}}$ (p) | $2 \rightarrow 1$ | 650 k | $2.65 \pm 0.05$ | $6.80 \pm 0.08$ | $10.78 \pm 0.13$ |
| $\mathrm{OR}, \mathrm{XNOR}_{\mathrm{AIL}}$ (d) | $1 \rightarrow 1$ | 438 k | $1.89 \pm 0.13$ | $5.12 \pm 0.21$ | $8.35 \pm 0.24$ |
| $\mathrm{OR}, \mathrm{AND}, \mathrm{XNOR}_{\mathrm{AIL}}$ (p) | $2 \rightarrow 1$ | 650 k | $\mathbf{2.67} \pm 0.10$ | $\mathbf{6.95} \pm 0.14$ | $\mathbf{10.96} \pm 0.16$ |

Since we could not flexibly scale the total number of parameters in the network with the original architecture, we modified the TMN architecture by adding an additional linear layer to the end of each module which projects from the activation function to the embedding space. The modules would otherwise terminate with an activation function, which makes it difficult to handle activation functions which map $2 \rightarrow 1$. We dub the modified version of the network "TMNx."

Comparing the TMN results in Table 8 to TMNx in Table 9, we can see that adding the extra linear layer improved performance of the network in of itself. Intuitively, this makes sense since the output of the TMN modules are weighted with the output of the gating network and then summed, and this weighting and summing of evidence is best performed with on logits instead of the truncated output of ReLU units. But also, performance may have improved just because the model became larger.

We performed a hyperparameter search across the training parameters for the new network against the validation set using the ReLU activation function only. We adopted the hyperparameters discovered for ReLU for all other activation functions. The batch size was reduced to 128 due to the increase in size of the model. The discovered hyperparameters were a learning rate of $3 \times 10^{-3}$ for both the module and gating network, and a weight decay of $1 \times 10^{-5}$. Other training hyperparameters, such as the ratio of negative samples to present, were left unchanged.

In order to match the number of parameters in the network, we used the original TMN hidden width of 64 for the module and gater networks with activation functions which map $1 \rightarrow 1$, and increased this to 70 for activation function which map $2 \rightarrow 1$, to maintain the total number of trainable parameters. To investigate a comprehensive set of baselines, we compared against every activation function implemented in PyTorch v1.10. The results are shown in Table 9.

We found that the ELU family of activations (ELU, CELU, SELU) performed best across all three top-k values. Our best activation functions were $\{\mathrm{OR}, \mathrm{AND}_{\mathrm{AIL}} \text{ (d)}\}$, followed by $\{\mathrm{OR}, \mathrm{AND}, \mathrm{XNOR}_{\mathrm{AIL}} \text{ (p)}\}$, which outperformed ReLU. The other AIL activation functions we

Table 9: Performance of TMNx networks at compositional zero-shot learning (CZSL) on the MIT-States dataset. Pre-activation width of 64 or 70 (depending on activation function, to keep the number of parameters approximately constant). Mean (standard error) of $n=5$ random initializations. Bold: best. Underlined: second best. Italic: no significant difference from best (two-sided Student's t-test, $p>0.05$). Background: color scale from second-worst in column to best, linear with accuracy value.

| Activation function | Mapping | Params | MIT-States Test Accuracy (%) | | |
|---|---|---|---|---|---|
| | | | Top-1 | Top-2 | Top-3 |
| ReLU | $1 \rightarrow 1$ | 1.15M | $2.76 \pm 0.08$ | $7.11 \pm 0.08$ | $11.26 \pm 0.09$ |
| LeakyReLU | $1 \rightarrow 1$ | 1.15M | $2.72 \pm 0.09$ | $6.99 \pm 0.17$ | $10.95 \pm 0.20$ |
| PReLU | $1 \rightarrow 1$ | 1.15M | $2.83 \pm 0.06$ | $7.25 \pm 0.14$ | $11.44 \pm 0.17$ |
| Softplus | $1 \rightarrow 1$ | 1.15M | $2.92 \pm 0.11$ | $7.46 \pm 0.19$ | $11.75 \pm 0.18$ |
| ELU | $1 \rightarrow 1$ | 1.15M | $3.13 \pm 0.05$ | $7.78 \pm 0.14$ | $12.04 \pm 0.20$ |
| CELU | $1 \rightarrow 1$ | 1.15M | $3.17 \pm 0.06$ | $7.81 \pm 0.15$ | $12.00 \pm 0.19$ |
| SELU | $1 \rightarrow 1$ | 1.15M | $3.05 \pm 0.04$ | $7.77 \pm 0.10$ | $12.37 \pm 0.16$ |
| GELU | $1 \rightarrow 1$ | 1.15M | $2.81 \pm 0.06$ | $7.19 \pm 0.11$ | $11.34 \pm 0.15$ |
| SiLU | $1 \rightarrow 1$ | 1.15M | $2.87 \pm 0.08$ | $7.34 \pm 0.14$ | $11.60 \pm 0.18$ |
| Hardswish | $1 \rightarrow 1$ | 1.15M | $2.84 \pm 0.08$ | $7.23 \pm 0.11$ | $11.48 \pm 0.11$ |
| Mish | $1 \rightarrow 1$ | 1.15M | $2.90 \pm 0.08$ | $7.37 \pm 0.17$ | $11.59 \pm 0.13$ |
| Softsign | $1 \rightarrow 1$ | 1.15M | $2.90 \pm 0.04$ | $7.28 \pm 0.09$ | $11.53 \pm 0.18$ |
| Tanh | $1 \rightarrow 1$ | 1.15M | $2.93 \pm 0.09$ | $7.47 \pm 0.16$ | $11.64 \pm 0.20$ |
| GLU | $2 \rightarrow 1$ | 1.16M | $2.59 \pm 0.06$ | $6.86 \pm 0.15$ | $10.91 \pm 0.24$ |
| Max | $2 \rightarrow 1$ | 1.16M | $2.45 \pm 0.08$ | $6.52 \pm 0.18$ | $10.53 \pm 0.29$ |
| Max, Min (d) | $1 \rightarrow 1$ | 1.15M | $2.53 \pm 0.10$ | $6.61 \pm 0.15$ | $10.65 \pm 0.15$ |
| XNOR$_{\text{AIL}}$ | $2 \rightarrow 1$ | 1.16M | $1.88 \pm 0.06$ | $4.94 \pm 0.15$ | $7.99 \pm 0.19$ |
| OR$_{\text{AIL}}$ | $2 \rightarrow 1$ | 1.16M | $2.61 \pm 0.08$ | $6.83 \pm 0.12$ | $10.97 \pm 0.20$ |
| OR, AND$_{\text{AIL}}$ (d) | $1 \rightarrow 1$ | 1.15M | $2.81 \pm 0.06$ | $7.12 \pm 0.10$ | $11.27 \pm 0.14$ |
| OR, XNOR$_{\text{AIL}}$ (p) | $2 \rightarrow 1$ | 1.16M | $2.61 \pm 0.08$ | $6.83 \pm 0.12$ | $10.97 \pm 0.20$ |
| OR, XNOR$_{\text{AIL}}$ (d) | $1 \rightarrow 1$ | 1.15M | $2.46 \pm 0.04$ | $6.51 \pm 0.11$ | $10.45 \pm 0.17$ |
| OR, AND, XNOR$_{\text{AIL}}$ (p) | $2 \rightarrow 1$ | 1.16M | $2.72 \pm 0.11$ | $7.15 \pm 0.19$ | $11.32 \pm 0.22$ |
| OR, AND, XNOR$_{\text{AIL}}$ (d) | $2 \rightarrow 3$ | 1.22M | $2.73 \pm 0.10$ | $6.95 \pm 0.16$ | $11.03 \pm 0.09$ |

considered peformed less well, and XNOR performed particularly poorly on this task. This suggests this is a domain where logical activation functions are less well suited.

