# OpenReview forum: "Logical Activation Functions: Logit-space equivalents of Boolean Operators"
_ICLR.cc/2022/Conference — ICLR 2022 Submitted_

### Official Review · Reviewer_FLuV · 2021-11-02

**Correctness:** 4
**Technical Novelty And Significance:** 4
**Empirical Novelty And Significance:** 3
**Recommendation:** 6
**Confidence:** 3

**Main Review:**

### Strengths
- The paper is clearly written.

- The core idea is novel and sounds.

- The paper provides strengths and weaknesses of the proposed approach with diverse experimental setup.

- The experiments include a good amount of tasks and data types from the image classification to transfer learning, abstract reasoning, and zero-shot learning on music, images, etc.

### Main questions and suggestions
1. Does "Linear Layer only" in Table 1, 5, 6 mean without any activation function?  In these tables, linear layers performs surprisingly well on some datasets with a very small amount of parameters (101k) compared to the models with the proposed activation functions (around 700k).
    1) Does this mean that the tasks/datasets are too simple? Or is there another reason that linear layers work reasonably well? Could authors comment on this phenomenon?
    2) Is there a reason to use a small model for the "linear layer only" model? Otherwise, similar parameter setup seems fair for comparison.

2. The performance of MaxOut is comparable to ${OR_{AIL}}$ for image-based tasks. Does this mean that most of the pre-activations become negative? Could authors provide more analysis between Maxout and $OR_{AIL}$? Plotting a maximum response histogram of the activations (similar to the one in the MaxOut networks paper (Goodfellow et al., 2013)) would be one way to compare their behaviors.

3. Experiments on more complex data like Imagenet would be useful to see how the proposed approach scales and whether the model is able to learn higher-order representation.

### Other suggestions
4. Authors mainly applied the proposed activation functions for MLP and CNN-based models. Can these activation functions be used for other common models like RNNs/GRU or transformer?

5. The related works listed in the paper seem limited. Adding previous works on proposing other activation functions would be helpful.

Minor:
Table 1 exceeds the paper size. The size needs to be reduced.


**Summary Of The Paper:**

The paper proposes new activation functions based on the approximation of AND, OR, XOR operations. In addition, ensembling strategies are proposed to combine features from these operators. The experiments on image classification, transfer learning, abstract reasoning, and zero-shot learning tasks show that the proposed activation functions perform reasonably well, especially OR for ResNet and an ensemble for zero-shot and transfer learning.

**Summary Of The Review:**

In general, the paper provides great idea, insights, and analysis. I added my major concerns in main questions and suggestions above. I am willing to increase my score once my concerns are addressed.

---------------------------------
### After rebuttal

I read the responses and other reviews. I generally like the idea of this paper and how it's written. However, my concerns are not fully addressed in the response, and I also agree with some of other reviewers' concerns. I keep my score.

---

> ### Author Response · Authors · 2021-11-22
> **Initial response to FLuV (1/2); transfer learning**
>
> We would like to thank FLuV for their positive review, and constructive feedback on the paper.
>
> > Does "Linear Layer only" in Table 1, 5, 6 mean without any activation function?
>
> Yes, that is correct. For "Linear Layer only", there is just one trainable weight matrix which maps from the ResNet-18 512d embedding space to the classification logits. We will be sure to clarify this.
>
> > In these tables, linear layers performs surprisingly well on some datasets with a very small amount of parameters (101k) compared to the models with the proposed activation functions (around 700k).
> > Does this mean that the tasks/datasets are too simple? Or is there another reason that linear layers work reasonably well? Could authors comment on this phenomenon?
>
> The pretrained network was trained on ImageNet. For transfer tasks which involve coarse-grained discrimination on images which are similar to ImageNet, the embedding generated by the pretrained model is already sufficient to separate the classes in the new dataset. Examples of this are Caltech101, where linear layer's 88.33 beats out using additional layers with non-linearities; and STL-10, where our best model (94.70) is on-par with the linear model (94.69).
> This does indeed mean these two transfer learning tasks are really too simple to be of interest to study. The performance is limited by the information retained by the pretrained embedding, but it appears that what task-relevant information is there is readily available with a linear layer without needing additional logic to interpret it.
>
> Other transfer learning tasks we attempted are less trivial and show a larger difference in performance across the activation functions, and a gap from the linear layer readout model.
> The Stanford Cars dataset involves fine-grained discrimination between different car models, for which features generated by a model pretrained on ImageNet are not effective (linear layer: 33.51; our best: 37.89).
> The SVHN dataset, which contains images of house numbers, is coarse-grained but uses images which are outside the domain of ImageNet, which makes the transfer-learning task more difficult (linear layer: 45.42; our best: 52.19).
>
> Please also refer to [our response to M59f](https://openreview.net/forum?id=Ck_iw4jMC4l&noteId=GMCWLEpBzv) for additional discussion on the choice of datasets and tasks used.
>
> > Is there a reason to use a small model for the "linear layer only" model? Otherwise, similar parameter setup seems fair for comparison.
>
> The number of parameters for the linear layer is equal to the size of the embedding space ($d=512$) multiplied by the number of classes ($10<c<196$) for the d-by-c weight matrix, plus the size of the bias term which equals the number of classes, i.e. $dc + c$.
> One could use fewer parameters, for instance by using a low rank matrix factorisation $W=UV$, where $U$ is a d-by-k matrix and $V$ is k-by-c. The information is compressed through the intermediate bottleneck of size $k$. This yields a total of $dk+kc$ trainable parameters for the weight matrix, which is smaller than $dc$ if $k<\text{min}(d,c)$.
> However, it is not really feasible to use more than $dc$ parameters. A series of linear layers without non-linearities in between them can be collapsed down into a single weight matrix. In this case, that matrix will be size d-by-c.

---

> ### Author Response · Authors · 2021-11-22
> **Initial response to FLuV (2/2)**
>
> > The performance of MaxOut is comparable to $OR_{AIL}$ for image-based tasks. Does this mean that most of the pre-activations become negative? Could authors provide more analysis between Maxout and OR\_AIL
>
> We don't think this is the case, as the final performance can be the same/similar even if the method used to get there is different.  However, we will indeed explore further to see if there are signs of $\text{OR}_\text{AIL}$ collapsing to Max or even to ReLU. Thank you for this suggestion.
>
>
> > Authors mainly applied the proposed activation functions for MLP and CNN-based models. Can these activation functions be used for other common models like RNNs/GRU or transformer?
>
> Our activation functions can be used in any domain.
> They can be readily applied to the output of a GRU or Transformer model, without having to modify the existing architecture.
> We were already planning on applying our logical activation functions to language understanding tasks next in this way, to see the performance when using the logical activation functions on high-level language representations. We think this is a compelling domain in which these activation functions may perform well.
>
> However, when FLuV asks about RNN/GRU, we get the impression they are asking whether we might be able to use these activation functions within the RNN cell itself (to compute probabilistic logical operations just before the multiplicative sigmoid gates, say). This is a very intriguing idea and one we will be interested to explore in future. We thank the reviewer for this suggestion.
>
> > The related works listed in the paper seem limited. Adding previous works on proposing other activation functions would be helpful.
>
> In our submission, we provide references for MaxOut, ELU, CELU, SELU, GELU, SiLU/Swish, Mish, but did not refer to all these activation functions by name, which we will address.
> We also have just realized that we did not actually provide a citation to early papers describing ReLU (e.g. Nair & Hinton 2010, Jarrett et al 2009, Fukushima 1980) --- that was a complete oversight and we will be sure to fix that.
> We would be delighted to include (and learn about!) any other relevant activation functions that any of the reviewers might suggest.

---

### Official Review · Reviewer_d99Z · 2021-11-02

**Correctness:** 3
**Technical Novelty And Significance:** 3
**Empirical Novelty And Significance:** 2
**Recommendation:** 5
**Confidence:** 4

**Main Review:**

The paper is technically sound and well-written, with some very minor comments on the latter.

In particular:
- You should list the activation functions you reference in the introduction by their names.
- Table 1 breaks the margins, this is unacceptable. The same holds true for some tables in the appendix.
- In section 2.4 you mention “For simplicity, we restrict ourselves to considering only k = 2, but note that these activation functions are generalisable to higher dimensions.”, and this is a perfectly valid argument. However you continue with “On the understanding that Goodfellow et al. (2013) found k = 2 yielded the best results for MaxOut networks, we anticipate the same will hold for the AIL activation functions.” Here I recommend you either drop that sentence, or provide more evidence for the conjecture, either via more experiments or a stronger argument of why you believe that is the case.
- Although Figs. 1-3 are great and help the reader understand better. Figs. 4-6 would benefit from improving the formatting. The label on Fig. 4, could be further improved to describe what we are seeing there. What do the colors mean? And in general for figs. 4-6 the numbers are too small.

On the content side, I do see the connection to MaxOut, however your activations are potentially better when they fall in the x+y regime, as the gradient would back-propagate on both sides, unlike MaxOut where you always only update either x or y. I believe this is potentially an advantage and could be mentioned in the paper. But at the same time, that makes me wonder whether it would behave the same as maxout for the k>2 experiments.

The biggest problem that I see with the paper is the weak empirical section when it comes to stacking your activation functions against other alternatives. There are many other alternatives and you even mention some of them in the introduction. However, you do not compare against them. Furthermore, the recent advances in learnable activation functions further improve the performance of DNNs beyond what ReLU can do.
Finally, I believe the work in [1] deserves at least some comments in your paper as related work.

[1] Godfrey, L. B., & Gashler, M. S. (2017, October). A parameterized activation function for learning fuzzy logic operations in deep neural networks. In 2017 IEEE International Conference on Systems, Man, and Cybernetics (SMC) (pp. 740-745). IEEE.


**Summary Of The Paper:**

In this paper presents a new set of activation functions based on a relaxation of the logical operators AND, OR, XNOR. This is motivated by the behavior of the operators under the logit space equivalence.
Moreover, the authors present approximations that are computationally more efficient based on simple max, min and addition operations.
As these binary operators are reduction operations, i.e., they convert two inputs into one, the authors note the similarity to the MaxOut and provide different alternatives on how to adapt the architectures to use the boolean activation functions.
Finally, the authors provide empirical results showing how the different potential combinations of logical activation functions behave for different datasets and architectures.

**Summary Of The Review:**

I like the activation functions introduced by the authors in this paper. They seem like a nice extension to MaxOut with semantical meaning on the operations they carry.

The weakness I see here are that the empirical section should compare against other activation functions and the missing related work.

---

> ### Author Response · Authors · 2021-11-10
> **Initial response to d99Z (1/2)**
>
> We thank reviewer d99Z for their considered feedback on the paper.
>
> We will address the minor corrections, thank you for pointing them out.
>
> > Table 1 breaks the margins, this is unacceptable. The same holds true for some tables in the appendix.
>
> With regards to Table 1 breaking the margins, our options to solve this are (1) scale down the table, (2) remove the standard error values, (3) rotate the table 90 degrees so it takes up a whole page and put it in the appendix.
> We think (1) would not be legible, and with regards to (2) we would like to keep the error values, hence we plan to relegate the table from the main text to the appendix (although we would be open to other suggestions).
>
> > Furthermore, the recent advances in learnable activation functions further improve the performance of DNNs beyond what ReLU can do. Finally, I believe the work in [1] deserves at least some comments in your paper as related work.
>
> Thank you for sharing this work with us, with which we were not previously familiar. It is most certainly related work and we will add it to the introduction section as relevant background material.
>
> > In section 2.4 you mention “For simplicity, we restrict ourselves to considering only k = 2, but note that these activation functions are generalisable to higher dimensions.”, and this is a perfectly valid argument. However you continue with “On the understanding that Goodfellow et al. (2013) found k = 2 yielded the best results for MaxOut networks, we anticipate the same will hold for the AIL activation functions.” Here I recommend you either drop that sentence, or provide more evidence for the conjecture, either via more experiments or a stronger argument of why you believe that is the case.
>
> Due to the similarity between our activation functions and MaxOut, and the fact that converting preactivation potentiations down to a compressed space with a $k \to 1$ activation function becomes quite parameter-expensive as $k$ is increased, we expected $k=2$ to be the best option.
> Nonetheless, we did in fact run some preliminary experiments with $k > 2$ on MNIST with MLP and CNN, which showed that $k = 2$ was indeed the preferred value, and we were not terribly surprised by this.
> These preliminary experiments were run with a different setup to our final results, and we decided not to explore $k > 2$ for the final experiments because we did not think the result was sufficiently interesting for us to include it in the paper.
> We felt that, with two/three novel activation functions being introduced plus ensembles of them also being considered, there was already a lot of things being introduced in the paper.
> However, we acknowledge that it is not appropriate for us to make assertions which are informed by preliminary work that is not part of the published article.
> We would be happy to rerun the $k > 2$ experiments to bring them into line with our current setup and include them in the paper as d99Z requests if the other reviewers and the AC are happy with us doing this.

---

> > ### Author Response · Authors · 2021-11-10
> > **Initial response to d99Z (2/2), Additional baselines**
> >
> > > The biggest problem that I see with the paper is the weak empirical section when it comes to stacking your activation functions against other alternatives. There are many other alternatives and you even mention some of them in the introduction. However, you do not compare against them.
> >
> > Over the course of our work on this project, we have run many of the experiments using other activation functions.
> > In particular, we used Swish/SiLU and PReLU as additional baselines.
> > For example, at a conference (without proceedings) last year, we presented our preliminary results which did include Swish.
> > An anonymised copy of the poster can be obtained from https://ufile.io/zti0r5w4 showing these results with Swish included.
> >
> > We decided to cut these additional baselines from the paper because it was not our primary focus --- we were trying to explore the concept of this new family of activation functions.
> > Our activation functions are a natural extension of ReLU to an additional dimension, and an extension of Max by changing the behaviour in only one quadrant, and thus they provide a unified way to see ReLU and Max as part of this family, but they are not directly related to the *ELU and Swish family of activation functions.
> > Hence, after considerable discussions in the weeks leading up to submission, we felt that comparison with *ELU activations would only be meaningful in the sense of "do we set a new SOTA/is this the best activation function", but not in the sense of exploring what our activation functions are adding.
> > Furthermore, we had not run all experiments with these additional baselines (due to computational demands), and wanted to be consistent with the activation functions run across our experiments.
> > We would be happy to add results on Swish/SiLU and PReLU back into the paper if the other reviewers and AC are happy with this.
> > After submitting the paper, we foresaw that lack of comparison with SOTA activation functions as a baseline could be a criticism of the paper (though as explained earlier, achieving SOTA was not a primary goal in itself as much as studying a new family of generally strongly performing functions), and we have in the meantime run additional experiments to gain results with Swish/SiLU and PReLU for the experiments where they were previously missing.
> >
> > Additionally, since submitting the paper, we have run some of our experiments against a very broad set of existing activation functions and we would be happy to share these results too if the reviewers and AC would like to see these as well to help put the results in context.

---

> > > ### Comment · Reviewer_d99Z · 2021-11-21
> > > **more development**
> > >
> > > I appreciate the comments and I understand the difficulties of introducing new activation functions. If your goal is not to introduce a new SOTA, then we need to find reasons why the activation function is interesting.
> > > In other words, why do we need logical activations? If they don’t bring more performance, what can they provide?
> > > Maybe they can be used to improve explainability in DNNs? Or maybe they can help train faster? Or what other reasons are there to use them?
> > > And if you are just characterizing them, you then definitely need more comparisons to other activations and other architectures.

---

> > ### Comment · Reviewer_d99Z · 2021-11-14
> > **Comments**
> >
> > I believe that you can solve the problems with the tables by just changing the names, e.g., OR_AIL to simply O, AND_AIL to A, etc. If that is not enough, then consider cutting away the number of parameters in the table, and just mention them in the paper or the appendix. It might even be enough to mention the scale of the parameters, e.g. ~600k, etc.
> >
> > My point regarding K=2, is that you should back up your claims or remove the claim. A reader doesn’t know what other experiments you might have run, but this is not a major issue in my opinion. The paper is not weaker if you just say you limit to the case of k=2 for simplicity as in maxout.

---

### Official Review · Reviewer_k3Hn · 2021-11-03

**Correctness:** 3
**Technical Novelty And Significance:** 2
**Empirical Novelty And Significance:** 2
**Recommendation:** 3
**Confidence:** 3

**Main Review:**

# Summary

This paper introduces three novel activation functions for neural networks, by modifying the logit-space versions of AND, OR, and XNOR in order to only use addition and comparison operations.  A large number of experiments show that in certain architectures and on certain tasks, some combinations of these activation functions deliver good performance.  While the results are somewhat interesting, I find the activation functions under-motivated (e.g. the motivations assume independence of input features) and worry that many of the experimental settings are designed to favor their new functions.  Similarly, the breadth of experiments means that each one is relatively under-reported in its details.  It would be helpful to have a more developed theory of these activation functions and more experiments in settings that are more standard.

Strengths:
	* Very large set of experiments
	* Novel design of new activation functions

Weaknesses:
	* New activations not sufficiently motivated (hinging on assuming independence of input dimensions)
	* _Too many_ experiments which are (a) insufficiently described and explained, and (b) may be biased towards favoring the new activations

# Minor comments

* p 4: "assuming inputs encode independent events": the motivations for the activations all depend on this independence assumption, which does not seem justified to me.  Can the authors explain why this is desirable?

* p 5: "access to all 16 boolean functions".  Since {and, or, not} is a complete basis for boolean functions, why not just look at an equivalent of not?

* Experimental details under-reported: e.g. a $p$-value is reported in 3.2, but it's unclear what the sample of results is (different random seeds?).  And in Section 3.5, it's unclear where to find the actual results supporting the claims in the second paragraph.  And in 3.6, there is merely a giant table of results, without any explanation of the results contained therein.


# Typographic comments

* p 2: "plus the the log" --> "plus the log"

* p 2: the sentence starting "If we suppose" doesn't have a consequent / main verb

**Summary Of The Paper:**

This paper introduces three novel activation functions for neural networks, by modifying the logit-space versions of AND, OR, and XNOR in order to only use addition and comparison operations.  A large number of experiments show that in certain architectures and on certain tasks, some combinations of these activation functions deliver good performance.  While the results are somewhat interesting, I find the activation functions under-motivated (e.g. the motivations assume independence of input features) and worry that many of the experimental settings are designed to favor their new functions.  Similarly, the breadth of experiments means that each one is relatively under-reported in its details.  It would be helpful to have a more developed theory of these activation functions and more experiments in settings that are more standard.

**Summary Of The Review:**

New activation functions are introduced and tested in a very wide range of experimental settings, but I find the motivations not sufficiently developed and the experimental results under-reported and motivated; more theoretical development would help assess this contribution.

---

> ### Author Response · Authors · 2021-11-16
> **Initial response to k3Hn's review (1/2)**
>
> > p 4: "assuming inputs encode independent events": the motivations for the activations all depend on this independence assumption, which does not seem justified to me. Can the authors explain why this is desirable?
>
> The derivations for the activation functions do indeed assume independence. This is not desirable --- but it is very pragmatic. If we did not make the assumption of independence, we would not be able to derive an equation for the probabilistic operations in a log-odds representation because we would need to know the feature correlation distribution $p(x|y)$.
> The assumption of feature independence is the same as that made in the Naive Bayes classifier. In data from the real world, features are never truly independent, yet Naive Bayes still has utility despite this. Similarly, we hope that the AIL activation functions have utility in doing approximately the "right" operation, even if it is not exactly correct.
>
> We explored whether the assumption of independence was true by running experiments described in Section 3.3 and Appendix A.9 (shown in Figure 17).
> We found that randomly selected pairs of preactivation features within the same layer have correlations that are given by a Gaussian-like distribution centered around zero.
> This was the case for all of the activation functions we tested.
> The behaviour of randomly selected pairs of features is thus reasonably consistent with the assumption of independence which we have made.
> We also investigated the correlation between the pairs of preactivation features which were passing into our two-dimensional activation functions.
> Here, we found the correlation structure is different, and the correlation depends on the activation function being used. With Max and $\text{OR}_\text{AIL}$ activations, the network learns to make the columns of the weight matrix (and hence the preactivation scores for the pair of features) be inversely correlated.
> With $\text{XNOR}_\text{AIL}$, the network learns features which are either positively or negatively correlated (a wider distribution of correlations than seen with random pairs of features).
> We observe that (in all cases) the network learns to make the features passed to the AIL activation functions be correlated instead of independent, despite our assumption of independence.
> So it appears clear that the assumption of independence is violated, but also that it doesn't *really* matter because the network is choosing to break the assumption and induce these correlations between the features to get better performance.
>
> > p 5: "access to all 16 boolean functions". Since {and, or, not} is a complete basis for boolean functions, why not just look at an equivalent of not?
>
> A logit-space equivalent of NOT is achieved by inverting the sign in the weight matrix. By including XNOR in addition to AND and OR, it allows the network to perform any of the boolean logic operations in a single layer, which would otherwise not have been possible.
>
> > a value is reported in 3.2, but it's unclear what the sample of results is (different random seeds?).
>
> Yes, the p-value is over different random seeds. The number of seeds ($n=10$) is reported in Appendix A.7, from when we had to relegate the corresponding figure (Fig. 16) into the appendix due to lack of space in the main text. We will clarify this in the main text, thank you for pointing it out.
>
> > in Section 3.5, it's unclear where to find the actual results supporting the claims in the second paragraph.
>
> The claims in Section 3.5 refer to Figure 6. We apologise for not including a reference to the figure which was being described.
>
> > in 3.6, there is merely a giant table of results, without any explanation of the results contained therein.
>
> We apologise for this oversight. The description of the results appears to have accidentally been omitted from the draft. We will address this error.

---

> ### Author Response · Authors · 2021-11-16
> **Initial response to k3Hn's review (2/2)**
>
> > [I] worry that many of the experimental settings are designed to favor their new functions.
>
> We hypothesised that these activation functions would perform well on tasks which involve logic reasoning. Consequently, we selected the tasks abstract reasoning, compositional zero-shot learning, and the Bach Chorales dataset to investigate this. Our other experiments (image classification on MNIST, CIFAR-10/100, transfer learning, and Covertype) are generic experiments. We do not attempt to claim that our activation functions are universally superior to others on all tasks, and did not intend to make such a claim. We will amend the text to make it clearer how we determined which experiments to run, and that these activation functions perform well on the tasks we tried them on but may not work in all cases.
>
> Additionally, we would like to make clear that for the abstract reasoning and compositional zero-shot learning tasks, we adapted pre-existing architectures that had been built by other researchers using ReLU, with hyperparameters optimized when using ReLU. Furthermore, in those experiments the networks containing our novel activation functions had fewer parameters. If anything, the experimental paradigm for these experiments is biased towards ReLU.
>
> Please see [our response to M59f](https://openreview.net/forum?id=Ck_iw4jMC4l&noteId=5_4V6BYAZm0) for a detailed description of our motivation for each of the datasets and tasks we selected for this study.
>
> > It would be helpful to have [...] more experiments in settings that are more standard.
>
> We would appreciate a little more direction from k3Hn as to which additional experiments they would like us to perform.
>
> > the breadth of experiments means that each one is relatively under-reported in its details
>
> Indeed, our eyes were bigger than our stomach and we performed more experiments than we had pages to cover. We apologise for this. In our opinion, once we have run an experiment it would be negligent not to report it (otherwise one can misreport the performance of a model through survivorship bias). We will relegate some experimental results to the appendix so we can describe all the results more clearly and thoroughly in the main text.

---

### Official Review · Reviewer_M59f · 2021-11-05

**Correctness:** 3
**Technical Novelty And Significance:** 2
**Empirical Novelty And Significance:** 2
**Recommendation:** 3
**Confidence:** 3

**Main Review:**

I do not think the paper and the idea is well motivated. The paper states that sigmoid is less applied and that ReLU is now applied more. But, the introduction does not explain why or make the connection to why we need these Boolean logic based activation functions. Some theoretical analysis would be helpful.

There is mention of biological networks, but it is not well cited by the paper.

In the first paragraph, I found these states somewhat loose:
- "an activation function is a non-linearity..."
- "this will convert the logits of features into probabilities."

Although several experiments were performed, I did not find the reason for why certain datasets & architectures were chosen as it relates to the presented activation functions. Also, why and when would the presented activation function perform better or worse than the used baselines?

In Table 1, I noticed it is interesting that the variance across (not +/- values themselves) the various activation configurations appear low (e.g., CIFAR10 concentrated around 82.x, STL-10 concentrated around 94.x).

In Figure 5, I was curious to know why the hyper-parameters were determined by random search. Was this uniformed? How does other popular methods (e.g., line search) affect the results?


**Summary Of The Paper:**

The paper develops and presents approximations of three Boolean logic based activation functions: AND, OR, XNOR. These activation functions are based on the principle that neurons encode logits to represent presence of features in the log-odds space (logit-space). The formulations are simple and straightforward. Empirical evaluation is conducted on several applications using the presented activations (under individual and ensemble configuration), where ReLU, max, min (with adjustments) are the competitive baselines.


**Summary Of The Review:**

While the methods presented are simple, the paper needs more work on motivating the methods presented and connecting it to the empirical evaluations.

---

> ### Author Response · Authors · 2021-11-16
> **Initial response to M59f's Review (Part 1)**
>
> We thank the reviewer for their helpful comments.
>
> > I do not think the paper and the idea is well motivated.
> > ...
> > why [do] we need these Boolean logic based activation functions. Some theoretical analysis would be helpful.
>
> The choice of activation functions and the rationale behind that choice are fundamentally important questions in the study of neural networks. Our motivation was to discover and systematically explore activation functions which follow naturally from considering pre-activations to be logits of features. Our objective was not to find a ”best” activation function, but to discover new ways to see activations (both existing ones, such as MaxOut, and our proposed ones) through the lenses of probabilistic representations and primitive logical operations. In our view, gaining new understanding of this sort is no less well-motivated than beating SOTA on a particular benchmark dataset, although we see that the way in which we originally presented this work may have given the impression that we were trying to achieve the latter; we appreciate the reviewer recognizing this, and we will adjust our presentation accordingly.
>
> Regarding theoretical analysis: we completely agree that such analysis can be helpful, and indeed that is why Section 2 is dedicated entirely to a theoretically- motivated derivation of the proposed family of activation functions. Further- more, we present this derivation because we also believe that it provides in- tuition about a probabilistic interpretation of these functions are doing, and the assumptions on which they are based. Beyond this, the natural theoretical questions that come to mind are either trivially obvious based on existing knowl- edge (e.g. the proposed activation functions can simulate ReLU and therefore they, too, allow for universal approximators with sufficient width etc.) or very challenging (e.g. defining the class of problems for which the proposed activa- tions are more effective, and articulating and proving that additional efficacy in principle). If the reviewer has particular theoretical questions in mind, we would of course be glad to engage in that discussion.
>
> > why and when would the presented activation function perform better or worse than the used baselines?
>
> This is of course a central question, and one we want to investigate more after setting out the foundations in the current paper. Our thinking has been as follows. Using activation functions equivalent to AND, OR, and XNOR in logit-space imposes a (soft) prior on the network. We anticipate that this prior is useful in situations where the network does indeed need to take the AND, OR, and XNOR (respectively) of the features. Intuitively, we anticipated that these activation functions would be most useful toward the end of the network, where one must learn to combine high-level features to generate a class, rather than feature extractors such as at the start of a CNN. For instance, a chair and a stool look quite different, but may need to be joined together in one class (the OR operator). A white elephant requires both the properties of being white and being an elephant (the AND operator). A picture of someone winking requires precisely one of their eyes be closed (the XOR operator).
>
> In contrast to this, low-level features in a CNN are more like Gabor filters that compare intensities of neigbouring pixels. We do not have a reason to presuppose that AND, OR, and XNOR are particularly useful manipulating such low-level features (though from our experiments, we found that using the AND or OR operator is actually quite effective in this setting). At some point in the middle of the network, we speculate that a CNN may benefit from doing logical operations between neigbouring spatial locations, or of features at the same spatial location, but would not be able to intuit where the transition would be.
>
> Additionally, inducing this prior on the network on how to manipulate the logits should be particularly helpful in situations where the model and dataset are smaller, since here one can not rely on the raw brute-strength of a very large model trained on a large dataset to be able to fit to the data. We say the prior is a soft prior because a $OR_{AIL}$ network clearly has the representational capacity to emulate ReLU throughout its structure. That said, in practice it is unlikely to do so because it is very expensive (effectively discarding half of the parameters in the network).
>
> We will gladly include such discussion in the paper; it is very much in line with our original motivations for formulating these activation functions.
>
> [continued in next part]

---

> > ### Author Response · Authors · 2021-11-16
> > **Initial response to M59f's Review (Part 2)**
> >
> > > There is mention of biological networks, but it is not well cited by the paper.
> >
> > Unfortunately, a more complete analogy with biological networks was cut from the paper in order to meet the 9 page limit. This may have removed the relevant citations. We will free up space in the paper by moving some experiments to the appendix, which will free up space to restore the lost text. The most pertinent reference is Gidon (2020), which demonstrated that an individual neuron can compute XOR by using dendritic non-linearities.
> >
> > *Albert Gidon, Timothy Adam Zolnik, Pawel Fidzinski, Felix Bold- uan, Athanasia Papoutsi, Panayiota Poirazi, Martin Holtkamp, Imre Vida, and Matthew Evan Larkum. Dendritic action potentials and computation in human layer 2/3 cortical neurons. Science, 367(6473):83–87, 2020.*
> >
> > >why certain datasets & architectures were chosen as it relates to the presented activation functions.
> >
> > We are grateful to the reviewer for asking this question, because it is both straightforward to address directly, and we believe adding these explanations to the paper will strengthen it considerably. Again, we will move some experiments to the Appendix to provide room for more context and detail to be provided for each experiment.
> > In addition to generic MLP networks, we looked for tasks which we thought might be involve doing logical reasoning on high-level features (as suggested by the above discussion).
> >
> > - The Covertype dataset was chosen as a dataset that contains only simple features (and not pixels of an image) on which we could study a simple MLP architecture. As we are not aware of a “standard” dataset to test on for such tasks, we chose this dataset based on its popularity on the UCI ML repository.
> >
> > - The Bach Chorales dataset was chosen because—in addition to being in continued use by ML researchers for decades—it presents an interesting op- portunity to consider a task where logical activation functions are intuitively applicable, as it is a relatively small dataset that has also been approached with rule-based frameworks, e.g. the expert system by Ebcioglu (1988). (And indeed, we found an advantage to using our activation functions in this context.)
> > - MNIST, CIFAR-10, CIFAR-100 are standard image datasets, commonly used, and we believe M59f is probably not objecting to us using them. We used a small MLP and CNN architectures for the experiments on MNIST so we could investigate the performance of the network for many configurations (varying the size of the network). We used ResNet-50, a very common deep convolutional architecture within the computer vision field, to evaluate the performance in the context of a deep network.
> > - The datasets used for the transfer learning task are all common and popular natural image datasets, with some containing coarse-grained classification (CIFAR-10), others fine-grained (Stanford Cars), and with a varying dataset size (5000–75000 training samples). We chose to do an experiment involving transfer learning because it is a common practical situation where one must train only a small network that handles high-level features, and is the sort of situation which involves manipulating high-level features, relying on the pretrained network to do the feature extraction.
> > - We considered other domains where logical reasoning is involved as a component of the task, and isolated abstract reasoning and compositional zero-shot learning as suitable tasks.
> > -- For abstract reasoning, we wanted to use an IQ style test, and determined that I-RAVEN was a state-of-the-art dataset within this domain (having fixed some problems with the pre-existing RAVEN dataset). We determined that the SRAN architecture from the paper which introduced I-RAVEN was still state- of-the-art on this task, and so used this. However, our choice of dataset was a toss-up between I-RAVEN and RAVEN-FAIR as the advantage of one against the other was not clear to us (both fixing seemingly the same issue with the precursor RAVEN dataset, but doing so in different ways). Had we selected the RAVEN-FAIR dataset, we would have used the MRNet architecture from the paper which introduced RAVEN-FAIR instead (which, similarly, is still SOTA on its new task).
> >
> > (continued in next part)

---

> > > ### Author Response · Authors · 2021-11-16
> > > **Initial response to M59f's Review (Part 3 of 3)**
> > >
> > > -- Another problem domain in which we thought it would be interesting to study our activation functions was compositional zero-shot learning (CZSL). This is because the task inherently involves combining an attribute with an object (i.e. the AND operation). For CZSL, we looked at SOTA methods on paperswithcode. The best performance was from SymNet, but this was only implemented in TensorFlow and our code was set up in pytorch so we built our experiments on the back of the second-best instead, which is the TMN architecture. In the TMN paper, they used two datasets: MIT-States and UT- Zappos-1. In our preliminary experiments, we found that the network started overfitting on MIT-States after around 6 epochs, but on UT-Zappos-1 it was overfitting after the first or second epoch (one can not tell beyond the fact the val performance is best for the first epoch). In the context of zero-shot learning, an epoch uses every image once, but there are also only a finite number of tasks in the dataset. Because there are multiple samples for each noun/adjective pair, and each noun only appears with a handful of adjectives and vice versa, there is in a way fewer tasks in one epoch than there are images. Hence it is possible for a zero-shot learning model to overfit to the training tasks in less than one epoch (recall that the network includes a pretrained ResNet model for extracting features from the images). We did not particularly want to labour with changing the way the experiments were run to introduce a validation loop part-way through an epoch in order to catch the model before it started overfit- ting because (this would break our job’s ability to resume training correctly if interrupted), or change the definition of an epoch to be each task once without repetition. Since we really already had more than enough experiments by this point, we just dropped UT-Zappos-1 and focused on MIT-States.
> > >
> > > > In Table 1, I noticed it is interesting that the variance across (not +/- values themselves) the various activation configurations appear low (e.g., CIFAR10 concentrated around 82.x, STL-10 concentrated around 94.x).
> > >
> > > Since the transfer learning experiments were performed without retraining the base network, performance is highly limited by the features which the base network produces, whose relevance depends on the other lap between the domain of the pretraining task (ImageNet) and the new task. It transpires that the features learned by a ResNet18 pre-trained on ImageNet are not very good for the fine-grained classification task of Stanford Cars, or the out-of-domain SVHN images. Meanwhile Caltech101 is a small dataset that is within the domain of ImageNet, and using only a linear read-out provides strong performance. Hence the variation between performance on datasets is much larger than the variation within datasets.
> > >
> > > > why the hyper-parameters were determined by random search. Was this uniformed? How does other popular methods (e.g., line search) affect the results?
> > >
> > > Details about the random search method used are provided in appendix A.9. The search was uniform, but done in three stages with the range of values narrowing at the end of each stage. The initial ranges were chosen to be wide such that the optimal configuration should be contained within them for all activation functions.
> > > We chose to do random search instead of grid search because it typically yields better results for the same number of test cases. Since the hyper-parameter search procedure was the most computationally expensive component of the pa- per, we only used one search technique and did not investigate the impact of the search method on the results. We found that the XNOR activation function required quite different hyper-parameters than the other activation functions, but OR and Max activation functions used similar hyper-parameters to ReLU. However, we have no reason to believe that the proposed AIL activation func- tions are more susceptible than others to the choice of hyper-parameters or the way in which hyper-parameters are selected. Thus, in terms of being affected by choice of hyper-parameter search, we believe our results are representative of what would be found with other methods as well. In early experiments, we conducted a variety hyper-parameter searches, and based on those results we felt that a better use of our computational resources (i.e. leading to more insight and understanding about the activation functions’ behaviour) was to in- stead apply them across a wide range of types of tasks and datasets (see above discussion), as we did in the present work.
> > >
> > > Again, we thank the reviewer for raising all of these points which we are very glad to address, and which will altogether strengthen the paper.

---

### Decision · Program_Chairs · 2022-01-20

**Decision:**

Reject

**Comment:**

The paper is well written and deals with a simple yet interesting introduction of approxiThe paper is well written and deals with a simple yet interesting introduction of approxiThe paper is well written and deals with a simple yet interesting introduction of approxiThe paper is well written and deals with a simple yet interesting introduction of approxiThe paper is well written and deals with a simple yet interesting introduction of approxiThe paper is well written and deals with a simple yet interesting introduction of approxiThe paper is well written and deals with a simple yet interesting introduction of approximate Boolean logic activation functions. A number of comparison experiments showed intriguing differences for problems with potential logical structures. Authors suggest a probabilistic rational/motivation, i.e., computation in logit-space, however, more theoretical investigation is critically needed to answer why they perform the way they do. There are a lot of activations in the literature, so perhaps it is not easy to make a distinct contribution in performance. Despite the large number of experiments the reviewers were not convinced on how they support authors claims and contributions. The reviewers and AC strongly encourage the authors to keep the direction and improve the paper for another conference.